# Deep Explicit Duration Switching Models
# for Time Series

**Abdul Fatir Ansari**[2][*][†]    **Konstantinos Benidis**[1][*]    **Richard Kurle**[1]    **Ali Caner Türkmen**[1]

**Harold Soh**[2]    **Alexander J. Smola**[1]    **Yuyang Wang**[1]    **Tim Januschowski**[1]

[1]Amazon Research    [2]National University of Singapore

Correspondence to: abdulfatir@u.nus.edu, {kbenidis, kurler}@amazon.com

## Abstract

Many complex time series can be effectively subdivided into distinct regimes that exhibit persistent dynamics. Discovering the switching behavior and the statistical patterns in these regimes is important for understanding the underlying dynamical system. We propose the Recurrent Explicit Duration Switching Dynamical System (RED-SDS), a flexible model that is capable of identifying both state- and time-dependent switching dynamics. State-dependent switching is enabled by a recurrent state-to-switch connection and an explicit duration count variable is used to improve the time-dependent switching behavior. We demonstrate how to perform efficient inference using a hybrid algorithm that approximates the posterior of the continuous states via an inference network and performs exact inference for the discrete switches and counts. The model is trained by maximizing a Monte Carlo lower bound of the marginal log-likelihood that can be computed efficiently as a byproduct of the inference routine. Empirical results on multiple datasets demonstrate that RED-SDS achieves considerable improvement in time series segmentation and competitive forecasting performance against the state of the art.

## 1   Introduction

Time series forecasting plays a key role in informing industrial and business decisions [17, 24, 8], while segmentation is useful for understanding biological and physical systems [40, 45, 34]. State Space Models (SSMs) [16] are a powerful tool for such tasks—especially when combined with neural networks [42, 12, 13]—since they provide a principled framework for time series modeling. One of the most popular SSMs is the Linear Dynamical System (LDS) [5, 43], which models the dynamics of the data using a continuous latent variable, called *state*, that evolves with Markovian linear transitions. The assumptions of LDS allow for exact inference of the states [27]; however, they are too restrictive for real-world systems that often exhibit piecewise linear or non-linear hidden dynamics with a finite number of operating modes or *regimes*. For example, the power consumption of a city may follow different hidden dynamics during weekdays and weekends. Such data are better explained by a Switching Dynamical System (SDS) [1, 21], an SSM with an additional set of latent variables called *switches* that define the operating mode active at the current timestep.

Switching events can be classified into time-dependent or state-dependent [33]. Historically, emphasis was placed on the former, which occurs after a certain amount of time has elapsed in a given regime. While in a vanilla SDS switch durations follow a geometric distribution, more complex long-term

---

[*]Equal contribution.

[†]Work done during an internship at Amazon Research.

35th Conference on Neural Information Processing Systems (NeurIPS 2021).

temporal patterns can be captured using explicit duration models [40, 9]. As a recent alternative to time-dependency, recurrent state-to-switch connections [35] have been proposed that capture state-dependent switching, i.e., a change that occurs when the state variable enters a region that is governed by a different regime. For added flexibility, these models can be used in conjunction with transition/emission distributions parameterized by neural networks [25, 19, 13, 30]. Recent works, e.g., [13, 30], proposed hybrid inference algorithms that exploit the graphical model structure to perform approximate inference for some latent variables and conditionally exact inference for others.

Despite these advances in representation and inference, modeling complex real-world temporal phenomena remains challenging. For example, state-of-the-art state-dependent models (e.g., [13]) lack the capacity to adequately capture time-dependent switching. Empirically, we find this hampers their ability to learn parsimonious segmentations when faced with complex patterns and long-term dependencies (see Fig. 1 for an example).

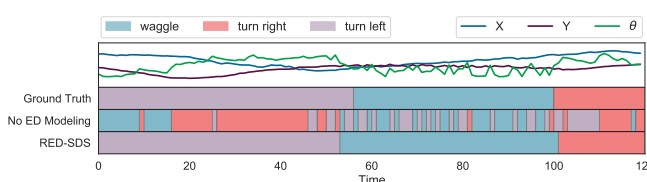

Figure 1: Segments (colored bars at the bottom) inferred by a baseline with no Explicit Duration (ED) modeling vs. our RED-SDS for a time series from the dancing bees dataset (top). The baseline struggles to learn long-term temporal patterns, particularly during the "waggle" phase of the bee dance.

Conversely, time-dependent switching models are "open-loop" and unable to model state-conditional behavioral transitions that are common in many systems, e.g., in autonomous or multi-agent systems [35]. Intuitively, the suitability of the switching model largely depends on the underlying data-generating process; city power consumption may be better modeled via time-dependent switching, whilst the motion of a ball bouncing between two walls is driven by its state. Indeed, complex real-world processes likely involve both types of switching behavior.

Motivated by this gap, we propose the Recurrent Explicit Duration Switching Dynamical System (RED-SDS) that captures *both* state-dependent and time-dependent switching. RED-SDS combines the recurrent state-to-switch connection with explicit duration models for switches. Notably, RED-SDS allows the incorporation of inductive biases via the hyperparameters of the duration models to better identify long-term temporal patterns. However, this combination also complicates inference, especially when using neural networks to model the underlying probability distributions. To address this technical challenge, we propose a hybrid algorithm that (i) uses an inference network for the continuous latent variables (states) and (ii) performs efficient exact inference for the discrete latent variables (switches and counts) using a forward-backward routine similar to Hidden Semi-Markov Models [48, 9]. The model is trained by maximizing a Monte Carlo lower bound of the marginal log-likelihood that can be efficiently computed by the inference routine.

We evaluated RED-SDS on two important tasks: segmentation and forecasting. Empirical results on segmentation show that RED-SDS is able to identify both state- and time-dependent switching patterns, considerably outperforming benchmark models. For example, Fig. 1 shows that RED-SDS addresses the oversegmentation that occurs with an existing strong baseline [13]. For forecasting, we illustrate the competitive performance of RED-SDS with an extensive evaluation against state-of-the-art models on multiple benchmark datasets. Further, we show how our model is able to simplify the forecasting problem by breaking the time series into different meaningful regimes without any imposed structure. As such, we manage to learn appropriate duration models for each regime and extrapolate the learned patterns into the forecast horizon consistently.

In summary, the key contributions of this paper are:

- RED-SDS, a novel non-linear state space model which combines the recurrent state-to-switch connection with explicit duration models to flexibly model switch durations;

- an efficient hybrid inference and learning algorithm that combines approximate inference for states with conditionally exact inference for switches and counts;

- a thorough evaluation on a number of benchmark datasets for time series segmentation and forecasting, demonstrating that RED-SDS can learn meaningful duration models, identify both state- and time-dependent switching patterns and extrapolate the learned patterns consistently into the future.

## 2 Background: switching dynamical systems

**Notation.** Matrices, vectors and scalars are denoted by uppercase bold, lowercase bold and lowercase normal letters, respectively. We denote the sequence $\{\mathbf{y}_1, \ldots, \mathbf{y}_T\}$ by $\mathbf{y}_{1:T}$, where $\mathbf{y}_t$ is the value of $\mathbf{y}$ at time $t$. In our notation, we do not further differentiate between random variables and their realizations.

Switching Dynamical Systems (SDS) are hybrid SSMs that use discrete "switching" states $z_t$ to index one of $K$ base dynamical systems with continuous states $\mathbf{x}_t$. The joint distribution factorizes as

$$p(\mathbf{y}_{1:T}, \mathbf{x}_{1:T}, z_{1:T}) = \prod_{t=1}^{T} p(\mathbf{y}_t|\mathbf{x}_t)p(\mathbf{x}_t|\mathbf{x}_{t-1}, z_t)p(z_t|z_{t-1}), \tag{1}$$

where $p(\mathbf{x}_1|\mathbf{x}_0, z_1)p(z_1|z_0) = p(\mathbf{x}_1|z_1)p(z_1)$ is the initial (continuous and discrete) state prior. The base dynamical systems have continuous state transition $p(\mathbf{x}_t|\mathbf{x}_{t-1}, z_t)$ and continuous or discrete emission $p(\mathbf{y}_t|\mathbf{x}_t)$ that can both be linear or non-linear.

The discrete transition $p(z_t|z_{t-1})$ of vanilla SDS is parametrized by a stochastic transition matrix $\mathbf{A} \in \mathbb{R}^{K \times K}$, where the entry $a_{ij} = \mathbf{A}(i, j)$ represents the probability of switching from state $i$ to state $j$. This results in an "open loop" as the transition only depends on the previous switch which inhibits the model from learning state-dependent switching patterns [35]. Further, the state duration (also known as the *sojourn time*) follows a geometric distribution [9], where the probability of staying in state $i$ for $d$ steps is $\rho_i(d) = (1 - a_{ii})a_{ii}^{d-1}$. This *memoryless* switching process results in frequent regime switching, limiting the ability to capture consistent long-term time-dependent switching patterns. In the following, we briefly discuss two approaches that have been proposed to improve the state-dependent and time-dependent switching capabilities in SDSs.

**Recurrent SDS.** Recurrent SDSs (e.g., [6, 35, 7, 30]) address state-dependent switching by changing the switch transition distribution to $p(z_t|\mathbf{x}_{t-1}, z_{t-1})$—called the state-to-switch recurrence—implying that the switch transition distribution changes at every step and the sojourn time no longer follows a geometric distribution. This extension complicates inference. Furthermore, the first-order Markovian recurrence does not adequately address long-term time-dependent switching.

**Explicit duration SDS.** Explicit duration SDSs are a family of models that introduce additional random variables to explicitly model the switch duration distribution. Explicit duration variables have been applied to both HMMs and SDSs with Gaussian linear continuous states; the resulting models are referred to as Hidden Semi-Markov Models (HSMMs) [38, 48], and Explicit Duration Switching Linear Gaussian SSMs (ED-SLGSSMs) [9, 40, 10], respectively. Several methods have been proposed in the literature for modeling the switch duration, e.g., using decreasing or increasing count, and duration-indicator variables. In the following, we briefly describe modeling switch duration using increasing count variables and refer the reader to Chiappa [9] for details.

Increasing count random variables $c_t$ represent the *run-length* of the currently active regime and can either increment by 1 or reset to 1. An increment indicates that the switch variable $z_t$ is copied over to the next timestep whereas a reset indicates a regular Markov transition using the transition matrix $\mathbf{A}$. Each of the $K$ switches has a distinct duration distribution $\rho_k$, a categorical distribution over $\{d_{\min}, \ldots, d_{\max}\}$, where $d_{\min}$ and $d_{\max}$ delimit the number of steps before making a Markov transition. Following [40, 9], the probability of a count increment is given by

$$v_k(c) = 1 - \frac{\rho_k(c)}{\sum_{d=c}^{d_{\max}} \rho_k(d)}. \tag{2}$$

The transition of count $c_t$ and switch $z_t$ variables is defined as

$$p(c_t|z_{t-1} = k, c_{t-1}) = \begin{cases} v_k(c_{t-1}) & \text{if } c_t = c_{t-1} + 1 \\ 1 - v_k(c_{t-1}) & \text{if } c_t = 1 \end{cases}, \tag{3}$$

$$p(z_t = j|z_{t-1} = i, c_t) = \begin{cases} \delta_{z_t=i} & \text{if } c_t > 1 \\ \mathbf{A}(i, j) & \text{if } c_t = 1 \end{cases}, \tag{4}$$

where $\delta_{\text{cond}}$ denotes the delta function which takes the value 1 only when $\text{cond}$ is true.

Although SDSs with explicit switch duration distributions can identify long-term time-dependent switching patterns, the switch transitions are not informed by the state—inhibiting their ability to

model state-dependent switching events. Furthermore, to the best of our knowledge, SDSs with explicit duration models have only been studied for Gaussian linear states [10, 9, 40].

## 3 Recurrent explicit duration switching dynamical systems

In this section we describe the Recurrent Explicit Duration Switching Dynamical System (RED-SDS) that combines both state-to-switch recurrence and explicit duration modeling for switches in a single non-linear model. We begin by formulating the generative model as a recurrent switching dynamical system that explicitly models the switch durations using increasing count variables. We then discuss how to perform efficient inference for different sets of latent variables. Finally, we discuss how to estimate the parameters of RED-SDS using maximum likelihood.

### 3.1 Model formulation

Consider the graphical model in Fig. 2 (a); the joint distribution of the counts $c_t \in \{1, \ldots, d_{\max}\}$, the switches $z_t \in \{1, \ldots, K\}$, the states $\mathbf{x}_t \in \mathbb{R}^m$, and the observations $\mathbf{y}_t \in \mathbb{R}^d$, conditioned on the control inputs $\mathbf{u}_t \in \mathbb{R}^c$, factorizes as

$$
\begin{aligned}
p_\theta(\mathbf{y}_{1:T}, \mathbf{x}_{1:T}, z_{1:T}, c_{1:T} | \mathbf{u}_{1:T}) = {} & p(\mathbf{y}_1|\mathbf{x}_1)p(\mathbf{x}_1|z_1, \mathbf{u}_1)p(z_1|\mathbf{u}_1) \\
& \cdot \left[ \prod_{t=2}^{T} p(\mathbf{y}_t|\mathbf{x}_t)p(\mathbf{x}_t|\mathbf{x}_{t-1}, z_t, \mathbf{u}_t)p(z_t|\mathbf{x}_{t-1}, z_{t-1}, c_t, \mathbf{u}_t)p(c_t|z_{t-1}, c_{t-1}, \mathbf{u}_t) \right].
\end{aligned} \tag{5}
$$

Similar to [40, 9], we consider increasing count variables $c_t$ to incorporate explicit switch durations into the model, i.e., $c_t$ can either increment by 1 or reset to 1 at every timestep and represent the run-length of the current regime. A self-transition is allowed after the exhaustion of $d_{\max}$ steps for flexibility. In the subsequent discussion we omit the control inputs $\mathbf{u}_t$ for clarity of exposition.

We model the initial prior distributions in Eq. (5) for the respective discrete and continuous case as

$$
p(z_1) = \operatorname{Cat}(z_1; \boldsymbol{\pi}), \tag{6}
$$

$$
p(\mathbf{x}_1|z_1) = \mathcal{N}(\mathbf{x}_1; \boldsymbol{\mu}_{z_1}, \boldsymbol{\Sigma}_{z_1}), \tag{7}
$$

where $\operatorname{Cat}$ denotes a categorical and $\mathcal{N}$ a multivariate Gaussian distribution. The transition distributions for the discrete variables (count and switch) are given by

$$
p(c_t|z_{t-1}, c_{t-1}) = \begin{cases} v_{z_{t-1}}(c_{t-1}) & \text{if} \quad c_t = c_{t-1} + 1 \\ 1 - v_{z_{t-1}}(c_{t-1}) & \text{if} \quad c_t = 1 \end{cases}, \tag{8}
$$

$$
p(z_t|\mathbf{x}_{t-1}, z_{t-1}, c_t) = \begin{cases} \delta_{z_t = z_{t-1}} & \text{if} \quad c_t > 1 \\ \operatorname{Cat}(z_t; \mathcal{S}_\tau(f_z(\mathbf{x}_{t-1}, z_{t-1}))) & \text{if} \quad c_t = 1 \end{cases}, \tag{9}
$$

where $\mathcal{S}_\tau$ is the tempered softmax function (cf. Section 3.3) with temperature $\tau$, and $f_z$ can be a linear function or a neural network. The probability of a count increment $v_k$ for a switch $k$ is defined via the duration model $\rho_k$ as in Eq. (2). The continuous state transition and the emission are given by

$$
p(\mathbf{x}_t|\mathbf{x}_{t-1}, z_t) = \mathcal{N}(\mathbf{x}_t; f_x^\mu(\mathbf{x}_{t-1}, z_t), f_x^\Sigma(\mathbf{x}_{t-1}, z_t)), \tag{10}
$$

$$
p(\mathbf{y}_t|\mathbf{x}_t) = \mathcal{N}(\mathbf{y}_t; f_y^\mu(\mathbf{x}_t), f_y^\Sigma(\mathbf{x}_t)), \tag{11}
$$

where $f_x^\mu, f_x^\Sigma, f_y^\mu, f_y^\Sigma$ are again linear functions or neural networks.

The model is general and flexible enough to handle both state- and time-dependent switching. The switch transitions $z_{t-1} \to z_t$ are conditioned on the previous state $\mathbf{x}_{t-1}$ which ensures that the switching events occur in a "closed loop". The switch duration models $\rho_k$ provide flexibility to stay long term in the same regime, allowing to better capture time-dependent switching. We use increasing count variables to incorporate switch durations into our model as they are more amenable to the case when the count transitions depend on the control $\mathbf{u}_t$. For instance, decreasing count variables, another popular option [11, 36, 9], *deterministically* count down from the sampled segment duration length to 1. This makes it difficult to condition the switch duration model on the control inputs. In contrast, increasing count variables increment or reset probabilistically at every timestep.

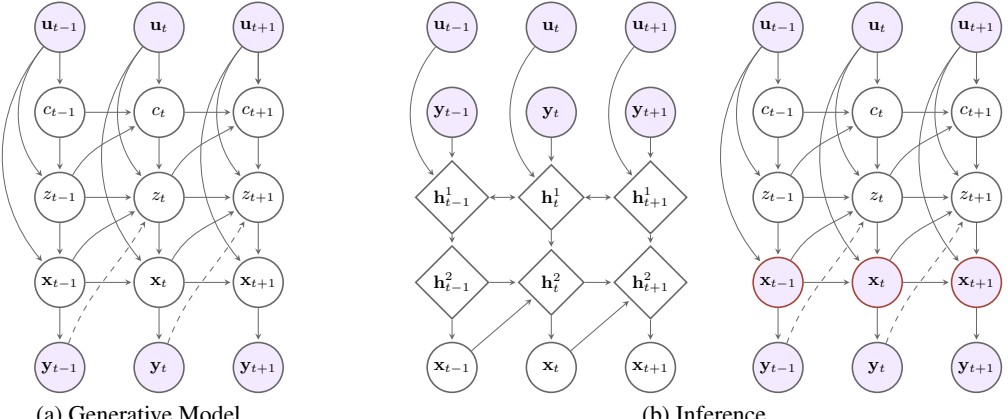

|             |             |
|:-----------:|:-----------:|
| (a) Generative Model | (b) Inference |

Figure 2: **(a)** Forward generative model of RED-SDS. **(b)** Left: Approximate inference for the states $\mathbf{x}_t$ using an inference network. $\mathbf{h}_t^1$ is given by a non-causal network and $\mathbf{h}_t^2$ is given by a causal RNN. Right: Exact inference for switch $z_t$ and count $c_t$ variables given pseudo-observations (highlighted in red) of $\mathbf{x}_t$ provided by the inference network. (Shaded) circles represent (observed) random variables, diamonds represent deterministic nodes, and dashed lines represent optional connections.

## 3.2 Inference

Exact inference is intractable in SDSs and scales exponentially with time [32]. Various approximate inference procedures have been developed for traditional SDSs [14, 21, 6], while more recently inference networks have been used for amortized inference for all or a subset of latent variables [25, 28, 13, 30]. Particularly, Dong et al. [13] used an inference network for the states and performed exact HMM-like inference for the switches, conditioned on the states. We take a similar approach and use an inference network for the continuous latent variables (states) and perform conditionally exact inference for the discrete latent variables (switches and counts) similar to the forward-backward procedure for HSMM [48, 9]. We define the variational approximation to the true posterior $p(\mathbf{x}_{1:T}, z_{1:T}, c_{1:T}|\mathbf{y}_{1:T})$ as $q(\mathbf{x}_{1:T}, z_{1:T}, c_{1:T}|\mathbf{y}_{1:T}) = q_\phi(\mathbf{x}_{1:T}|\mathbf{y}_{1:T})p_\theta(z_{1:T}, c_{1:T}|\mathbf{y}_{1:T}, \mathbf{x}_{1:T})$ where $\phi$ and $\theta$ denote the parameters of the inference network and the generative model respectively.

**Approximate inference for states.** The posterior distribution of the states, $q_\phi(\mathbf{x}_{1:T}|\mathbf{y}_{1:T})$, is approximated using an inference network. We first process the observation sequence $\mathbf{y}_{1:T}$ using a non-causal network such as a bi-RNN or a Transformer [46] to simulate smoothing by incorporating both past and future information. The non-causal network returns an embedding of the data $\mathbf{h}_{1:T}^1$ which is then fed to a causal RNN that outputs the posterior distribution $q_\phi(\mathbf{x}_{1:T}|\mathbf{y}_{1:T}) = \prod_t q(\mathbf{x}_t|\mathbf{x}_{1:t-1}, \mathbf{h}_{1:T}^1)$. See Fig. 2 (b) for an illustration of the inference procedure.

**Exact inference for counts and switches.** Inference for the switches $z_{1:T}$ and the counts $c_{1:T}$ can be performed exactly conditioned on states $\mathbf{x}_{1:T}$ and observations $\mathbf{y}_{1:T}$. Samples from the approximate posterior $\tilde{\mathbf{x}}_{1:T} \sim q(\mathbf{x}_{1:T}|\mathbf{y}_{1:T})$ are used as pseudo-observations of $\mathbf{x}_{1:T}$ to infer the posterior distribution $p_\theta(z_{1:T}, c_{1:T}|\mathbf{y}_{1:T}, \tilde{\mathbf{x}}_{1:T})$. A naive approach to infer this distribution is by treating the pair $(c_t, z_t)$ as a "meta switch" that takes $Kd_{\max}$ possibles values and perform HMM-like forward-backward inference. However, this results in a computationally expensive $O(TK^2d_{\max}^2)$ procedure that scales poorly with $d_{\max}$. Fortunately, we can pre-compute some terms in the forward-backward equations by exploiting the fact that the count variable can only increment by 1 or reset to 1 at every timestep. This results in an $O(TK(K + d_{\max}))$ algorithm that scales gracefully with $d_{\max}$ [9]. The forward $\alpha_t$ and backward $\beta_t$ variables, defined as

$$\alpha_t(z_t, c_t) = p(\mathbf{y}_{1:t}, \mathbf{x}_{1:t}, z_t, c_t), \tag{12}$$
$$\beta_t(z_t, c_t) = p(\mathbf{y}_{t+1:T}, \mathbf{x}_{t+1:T}|\mathbf{x}_t, z_t, c_t), \tag{13}$$

can be computed by modifying the forward-backward recursions used for the HSMM [9] to handle the additional observed variables $\mathbf{x}_{1:t}$. We refer the reader to Appendix A.1 for the exact derivation.

### 3.3 Learning

The parameters $\{\phi, \theta\}$ can be learned by maximizing the evidence lower bound (ELBO):

$$
\begin{aligned}
\mathcal{L}_{\text{ELBO}} &= \mathbb{E}_{q(\mathbf{x}_{1:T}|\mathbf{y}_{1:T})p(z_{1:T},c_{1:T}|\mathbf{y}_{1:T},\mathbf{x}_{1:T})} \left[ \log \frac{p(\mathbf{y}_{1:T}, \mathbf{x}_{1:T}, z_{1:T}, c_{1:T},)}{q(\mathbf{x}_{1:T}|\mathbf{y}_{1:T})p(z_{1:T}, c_{1:T}|\mathbf{y}_{1:T}, \mathbf{x}_{1:T})} \right] \\
&= \mathbb{E}_{q(\mathbf{x}_{1:T}|\mathbf{y}_{1:T})} \left[ \log \frac{p(\mathbf{y}_{1:T}, \mathbf{x}_{1:T})}{q(\mathbf{x}_{1:T}|\mathbf{y}_{1:T})} \right].
\end{aligned}
\tag{14}
$$

The likelihood term $p(\mathbf{y}_{1:T}, \mathbf{x}_{1:T})$ can be computed using the forward variable $\alpha_T(z_T, c_T)$ by marginalizing out the switches and the counts,

$$
p(\mathbf{y}_{1:T}, \mathbf{x}_{1:T}) = \sum_{z_T, c_T} \alpha_T(z_T, c_T),
\tag{15}
$$

and the entropy term $-\mathbb{E}_{q(\mathbf{x}_{1:T}|\mathbf{y}_{1:T})}\left[\log q(\mathbf{x}_{1:T}|\mathbf{y}_{1:T})\right]$ can be computed using the approximate posterior $q(\mathbf{x}_{1:T}|\mathbf{y}_{1:T})$ output by the inference network. The ELBO can be maximized via stochastic gradient ascent given that the posterior $q(\mathbf{x}_{1:T}|\mathbf{y}_{1:T})$ is reparameterizable.

We note that Dong et al. [13] used a lower bound for the likelihood term in Switching Non-Linear Dynamical Systems (SNLDS); however, it can be computed succinctly by marginalizing out the discrete random variable (i.e., the switch in SNLDS) from the forward variable $\alpha_T$, similar to Eq. (15). Using our objective function, we observed that the model was less prone to posterior collapse (where the model ends up using only one switch) and we did not require the additional ad-hoc KL regularizer used in Dong et al. [13]. Please refer to Appendix B.4 for a brief discussion on the likelihood term in SNLDS.

**Temperature annealing.** We use the tempered softmax function $\mathcal{S}_\tau$ to map the logits to probabilities for the switch transition $p(z_t|\mathbf{x}_{t-1}, z_{t-1}, c_t = 1)$ and the duration models $\rho_k(d)$ which is defined as

$$
\mathcal{S}_\tau(\mathbf{o})_i = \frac{\exp\left(\frac{o_i}{\tau}\right)}{\sum_j \exp\left(\frac{o_j}{\tau}\right)},
\tag{16}
$$

where $\mathbf{o}$ is a vector of logits. The temperature $\tau$ is deterministically annealed from a high value during training. The initial high temperature values soften the categorical distribution and encourage the model to explore all switches and durations. This prevents the model from getting stuck in poor local minima that ignore certain switches or longer durations which might explain the data better.

## 4 Related work

The most relevant components of RED-SDS are recurrent state-to-switch connections and the explicit duration model, enabling both for state- and time-dependent switching. Additionally, RED-SDS allows for efficient approximate inference (analytic for switches and counts), despite parameterizing the various conditional distributions through neural networks. Existing methods address only a subset of these features as we discuss in the following.

The most prominent SDS is the Switching Linear Dynamical System (SLDS), where each regime is described by linear dynamics and additive Gaussian noise. A major focus of previous work has been on efficient approximate inference algorithms that exploit the Gaussian linear substructure (e.g., [21, 49, 14]). In contrast to RED-SDS, these models lack recurrent state-to-switch connections and duration variables and are limited to linear regimes.

Previous work has addressed the state-dependent switching by introducing a connection to the continuous state of the dynamical system [6, 35, 7, 30]. The additional recurrence complicates inference w.r.t. the continuous states; prior work uses expensive sampling methods in order to approximate the corresponding integrals [6] or as part of a message passing algorithm for joint inference of states and parameters [35]. On the other hand, ARSGLS [30] avoids sampling the continuous states by using conditionally linear state-to-switch connections and softmax-transformed Gaussian switch variables. However, both the ARSGLS and the related KVAE [19] can be interpreted as an SLDS with "soft" switches that interpolate linear regimes continuously rather than truly discrete states. This makes them less suited for time series segmentation compared to RED-SDS. Contrary to

the aforementioned models, RED-SDS allows non-linear regimes described by neural networks and incorporates a discrete explicit duration model without complicating inference w.r.t. the continuous states, since closed-form expressions are used for the discrete variables instead. Using amortized variational inference for continuous variables and analytic expressions for discrete variables has been proposed previously for segmentation in SNLDS [13]. RED-SDS extends this via an additional explicit duration variable that represents the run-length for the currently active regime.

Explicit duration variables have previously been proposed for changepoint detection [2, 3] and segmentation [10, 26]. For instance, BOCPD [2] is a Bayesian online changepoint detection model with explicit duration modeling. RED-SDS improves upon BOCPD by allowing for segment labeling rather than just detecting changepoints. The HDP-HSMM [26] is a Bayesian non-parametric extension to the traditional HSMM. Recent work [11, 36] has also combined HSMM with RNNs for amortized inference. These models—being variants of HSMM—do not model the latent dynamics of the data like RED-SDS. Chiappa and Peters [10] proposed approximate inference techniques for a variant of SLDS with explicit duration modeling. In contrast, RED-SDS is a more general non-linear model that allows for efficient amortized inference—closed-form w.r.t. the discrete latent variables.

## 5   Experiments

In this section, we present empirical results on two prominent time series tasks: segmentation and forecasting. Our primary goals were to determine if RED-SDS (a) can discover meaningful switching patterns in the data in an unsupervised manner, and (b) can probabilistically extrapolate a sequence of observations, serving as a viable generative model for forecasting. In the following, we discuss the main results and relegate details to the appendix.

### 5.1   Segmentation

We experimented with two instantiations of our model: `RED-SDS` (complete model) and `ED-SDS`, the ablated variant without state-to-switch recurrence. We compared against the closely related `SNLDS` [13] trained with a modified objective function. The original objective proposed in [13] suffered from training difficulties: it resulted in frequent posterior collapse and was sensitive to the cross-entropy regularization term. Our version of `SNLDS` can be seen as a special case of `RED-SDS` with $d_{\max} = 1$, i.e., without the explicit duration modeling (cf. Appendix B.4). We also conducted preliminary experiments on soft-switching models: `KVAE` [19] and `ARSGLS` [30]. However, these models use a continuous interpolation of the different operating modes which cannot always be correctly assigned to a single discrete mode, hence we do not report these unfavorable findings here (cf. Appendix B.4). For all models, we performed segmentation by taking the most likely value of the switch at

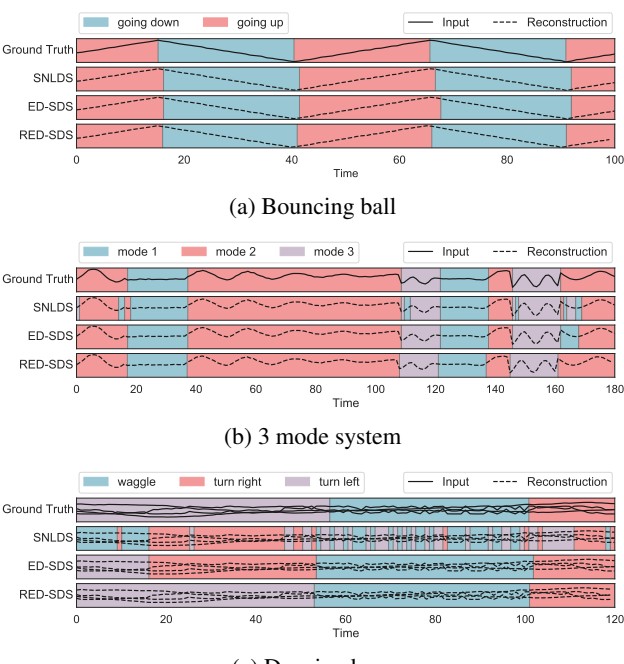

(a) Bouncing ball

(b) 3 mode system

(c) Dancing bees

Figure 3: Qualitative segmentation results on the bouncing ball, 3 mode system, and dancing bees datasets. Background colors represent the different operating modes.

each timestep from the posterior distribution over the switches. As the segmentation labels are arbitrary and may not match the ground truth labels, we evaluated the models using multiple metrics: frame-wise segmentation accuracy (after matching the labelings using the Hungarian algorithm [29]), Normalized Mutual Information (NMI) [47], and Adjusted Rand Index (ARI) [23] (cf. Appendix B.2).

Table 1: Quantitative results on segmentation tasks. Accuracy, NMI, and ARI denote the frame-wise segmentation accuracy, the Normalized Mutual Information, and the Adjusted Rand Index metrics respectively (higher values are better). Mean and standard deviation are computed over 3 independent runs.

| | | bouncing ball | 3 mode system | dancing bees | dancing bees(K=2) |
|---|---|---|---|---|---|
| Accuracy | SNLDS | **0.97±0.00** | 0.82±0.08 | 0.44±0.01 | 0.63±0.02 |
| | ED-SDS (ours) | 0.95±0.00 | 0.97±0.00 | 0.56±0.06 | 0.79±0.09 |
| | RED-SDS (ours) | **0.97±0.00** | **0.98±0.00** | **0.73±0.10** | **0.91±0.04** |
| NMI | SNLDS | **0.83±0.01** | 0.63±0.08 | 0.10±0.04 | 0.05±0.02 |
| | ED-SDS (ours) | 0.71±0.00 | 0.89±0.01 | 0.28±0.02 | 0.31±0.17 |
| | RED-SDS (ours) | 0.81±0.00 | **0.91±0.01** | **0.48±0.07** | **0.60±0.09** |
| ARI | SNLDS | **0.90±0.01** | 0.67±0.11 | 0.10±0.03 | 0.07±0.02 |
| | ED-SDS (ours) | 0.81±0.01 | 0.93±0.00 | 0.27±0.04 | 0.36±0.19 |
| | RED-SDS (ours) | 0.88±0.00 | **0.95±0.01** | **0.53±0.11** | **0.68±0.11** |

We conducted experiments on three benchmark datasets: bouncing ball, 3 mode system, and dancing bees to investigate different segmentation capabilities of the models. We refer the reader to Appendix B.1 for details on how these datasets were generated/preprocessed. For all the datasets, we set the number of switches equal to the number of ground truth operating modes.

**Bouncing ball.** We generated the bouncing ball dataset similar to [13], which comprises univariate time series that encode the location of a ball bouncing between two fixed walls with a constant velocity and elastic collisions. The underlying system switches between two operating modes (going up/down) and the switching events are completely governed by the state of the ball, i.e., a switch occurs only when the ball hits a wall. As such, the switching events are best explained by state-to-switch recurrence. All models are able to segment this simple dataset well as shown qualitatively in Fig 3 (a) and quantitatively in Table 1. We note that despite the seemingly qualitative equivalence, models with state-to-switch recurrence perform best quantitatively. RED-SDS learns to ignore the duration variable by assigning almost all probability mass to shorter durations (cf. Appendix B.5), which is intuitive since the recurrence best explains this dataset.

**3 mode system.** We generated this dataset from a switching linear dynamical system with 3 operating modes and an explicit duration model for each mode (shown in Fig. 4 (a)). We study this dataset in the context of time-dependent switching—the operating mode switches after a specific amount of time elapses based on its duration model. Both ED-SDS and RED-SDS learn to segment this dataset almost perfectly as shown in Fig. 3 (b) and Table 1 owing to their ability to explicitly model switch durations. In contrast, SNLDS fails to completely capture the long-term temporal patterns, resulting in spurious short-term segments as shown in Fig. 3 (b). Moreover, RED-SDS is able to recover the duration models associated with the different modes (Fig. 4). These results demonstrate that explicit duration models can better identify the time-dependent switching patterns in the data and can leverage prior knowledge about the switch durations imparted via the $d_{\min}$ and $d_{\max}$ hyperparameters.

**Dancing bees.** We used the publicly-available dancing bees dataset [40]—a challenging dataset that exhibits long-term temporal patterns and has been studied previously in the context of time series segmentation [41, 39, 18]. The dataset comprises trajectories of six dancer honey bees performing the waggle dance. Each trajectory consists of the 2D coordinates and the heading angle of a bee at every timestep with three possible types of motion: waggle, turn right, and turn left. Fig. 3 (c) shows that RED-SDS is able to segment the complex long-term motion patterns quite well. In contrast, ED-SDS identifies the long segment durations but often infers the mode inaccurately while SNLDS strug-

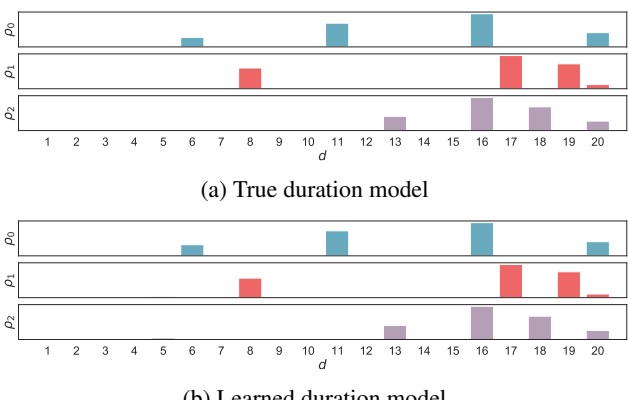

(a) True duration model

(b) Learned duration model

Figure 4: The ground truth duration model for the 3 mode system dataset (top) and the duration model learned by RED-SDS (bottom). The x-axis represents the durations from 1 to 20 and the y-axis represents the duration probabilities of the 3 modes $\rho_0(d), \rho_1(d)$, and $\rho_2(d)$.

Table 2: CRPS metrics (lower is better). Mean and standard deviation are computed over 3 independent runs. The method achieving the best result is highlighted in **bold**.

|            | exchange | solar | electricity | traffic | wiki |
|------------|----------|-------|-------------|---------|------|
| DeepAR     | 0.019±0.002 | 0.440±0.004 | **0.062±0.004** | 0.138±0.001 | 0.855±0.552 |
| DeepState  | 0.017±0.002 | 0.379±0.002 | 0.088±0.007 | 0.131±0.005 | 0.338±0.017 |
| KVAE-MC    | 0.020±0.001 | 0.389±0.005 | 0.318±0.011 | 0.261±0.016 | 0.341±0.032 |
| KVAE-RB    | 0.018±0.001 | 0.393±0.006 | 0.305±0.022 | 0.221±0.002 | 0.317±0.013 |
| RSGLS-ISSM | 0.014±0.001 | **0.358±0.001** | 0.091±0.004 | 0.206±0.002 | 0.345±0.010 |
| ARSGLS     | 0.022±0.001 | 0.371±0.007 | 0.154±0.005 | 0.175±0.008 | **0.283±0.006** |
| RED-SDS (ours) | **0.013±0.001** | 0.419±0.010 | 0.066±0.002 | **0.129±0.002** | 0.318±0.006 |

gles to learn the long-term motion patterns resulting in oversegmentation. This limitation of SNLDS is particularly apparent in the "waggle" phase of the dance which involves rapid, shaky motion. We also observed that sometimes ED-SDS and RED-SDS combined the turn right and turn left motions into a single switch, effectively segmenting the time series into regular (turn right and turn left) and waggle motion. This results in another reasonable segmentation, particularly in the absence of ground-truth supervision. We thus reevaluated the results after combining the turn right and turn left labels into a single label and present these results under dancing bees(K=2) in Table 1. Empirically, RED-SDS significantly outperforms ED-SDS and SNLDS on both labelings of the dataset. This suggests that real-world phenomena are better modeled by a combination of state- and time-dependent modeling capacities via state-to-switch recurrence and explicit durations, respectively.

## 5.2 Forecasting

We evaluated RED-SDS in the context of time series forecasting on 5 popular public datasets available in GluonTS [4], following the experimental set up of [30]. The datasets have either hourly or daily frequency with various seasonality patterns such as daily, weekly, or composite. In Appendix C.1 we provide a detailed description of the datasets. We compared RED-SDS to closely related forecasting models: ARSGLS and its variant RSGLS-ISSM [30]; KVAE-MC and KVAE-RB, which refer to the original KVAE [19] and its Rao-Blackwellized variant (as described in [30]) respectively; DeepState [42]; and DeepAR [44], a strong *discriminative* baseline that uses an autoregressive RNN (cf. Appendix C.4 for a discussion on these baselines).

We used data prior to a fixed forecast date for training and test the forecasts on the remaining unseen data; the probabilistic forecasts are conditioned on the training range and computed with 100 samples for each method. We used a forecast window of 150 days and 168 hours for datasets with daily and hourly frequency, respectively. We evaluated the forecasts using the *continuous ranked probability score* (CRPS) [37], a proper scoring rule [22] (cf. Appendix C.2). The results are reported in Table 2; RED-SDS compares favorably or competitively to the baselines on 4 out of 5 datasets.

Figure 5 illustrates how RED-SDS can infer meaningful switching patterns from the data and extrapolate the learned patterns into the future. It perfectly reconstructs the past of the time series and segments it in an interpretable manner without an imposed

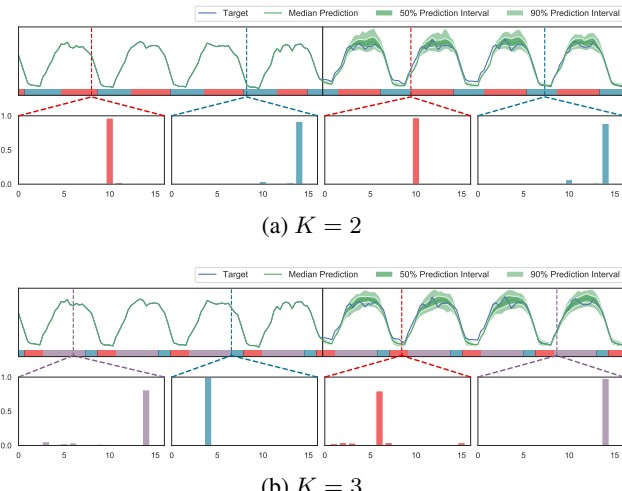

(a) $K = 2$

(b) $K = 3$

Figure 5: Segmentation and forecasting on an electricity time series for (a) $K = 2$ and (b) $K = 3$ switches. The black vertical line indicates the start of forecasting. The plots at the second row of each figure indicate the duration model at the timestep marked by the corresponding vertical dashed lines.

seasonality structure, e.g., as used in `DeepState` and `RSGLS-ISSM`. The same switching pattern is consistently predicted into the future, simplifying the forecasting problem by breaking the time series into different regimes with corresponding properties such as trend or noise variance. Further, the duration models at several timesteps (the duration model is conditioned on the control $\mathbf{u}_t$) indicate that the model has learned how long each regime lasts and therefore avoids oversegmentation which would harm the efficient modeling of each segment. Notably, the model learns meaningful regime durations that sum up to the 24-hour day/night period for both $K = 2$ and $K = 3$ switches. Thus, `RED-SDS` brings the added benefit of interpretability—both in terms of the discrete operating mode and the segment durations—while obtaining competitive quantitative performance relative to the baselines.

## 6 Conclusion and future work

Many real-world time series exhibit prolonged regimes of consistent dynamics as well as persistent statistical properties for the durations of these regimes. By explicitly modeling both state- and time-dependent switching dynamics, our proposed RED-SDS can more accurately model such data. Experiments on a variety of datasets show that RED-SDS—when equipped with an efficient inference algorithm that combines amortized variational inference with exact inference for continuous and discrete latent variables—improves upon existing models on segmentation tasks, while performing similarly to strong baselines for forecasting.

One current challenge of the proposed model is that learning interpretable segmentation sometimes requires careful hyperparameter tuning (e.g., $d_{\min}$ and $d_{\max}$). This is not surprising given the flexible nature of the neural networks used as components in the base dynamical system. A promising future research direction is to incorporate simpler models that have a predefined structure, thus exploiting domain knowledge. For instance, many forecasting models such as DeepState and RSGLS-ISSM parametrize classical level-trend and seasonality models in a non-linear fashion. Similarly, simple forecasting models with such structure could be used as base dynamical systems along with more flexible neural networks. Another interesting application is semi-supervised time series segmentation. For timesteps where the correct regime label is known, it is straightforward to condition on this additional information rather than performing inference; this may improve segmentation accuracy while providing an inductive bias that corresponds to an interpretable segmentation.

## Funding disclosure

This work was funded by Amazon Research. This research is supported in part by the National Research Foundation Singapore under its AI Singapore Programme (Award Number: AISG-RP-2019-011) to H. Soh.

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
