# A  Model: additional details

## A.1  Forward-backward algorithm

As mentioned in Section 3.2, inference for the discrete latent variables, i.e., the counts $c_{1:T}$ and the switches $z_{1:T}$, can be performed exactly conditioned on states $\mathbf{x}_{1:T}$ and observations $\mathbf{y}_{1:T}$. We first define the forward $\alpha_t$ and backward $\beta_t$ variables as

$$\alpha_t(z_t, c_t) = p(\mathbf{y}_{1:t}, \mathbf{x}_{1:t}, z_t, c_t), \tag{17}$$
$$\beta_t(z_t, c_t) = p(\mathbf{y}_{t+1:T}, \mathbf{x}_{t+1:T}|\mathbf{x}_t, z_t, c_t). \tag{18}$$

The smoothed posterior over the switches and counts, $p(z_t, c_t|\mathbf{y}_{1:T}, \mathbf{x}_{1:T})$, can be computed using $\alpha_t$ and $\beta_t$ as

$$
\begin{aligned}
\gamma_t(z_t, c_t) = p(z_t, c_t|\mathbf{y}_{1:T}, \mathbf{x}_{1:T}) &\propto p(\mathbf{y}_{1:T}, \mathbf{x}_{1:T}, z_t, c_t) \\
&= p(\mathbf{y}_{t+1:T}, \mathbf{x}_{t+1:T}|\mathbf{x}_t, z_t, c_t)p(\mathbf{y}_{1:t}, \mathbf{x}_{1:t}, z_t, c_t) \\
&= \beta_t(z_t, c_t)\alpha_t(z_t, c_t).
\end{aligned} \tag{19}
$$

In the following, we derive the recursions for $\alpha_t$ and $\beta_t$, similar to the forward-backward algorithm for HSMMs [9, Chapter 3]:

$$
\begin{aligned}
\alpha_1(z_1, c_1) &= \delta_{c_1=1}p(\mathbf{y}_1, \mathbf{x}_1|z_1)p(z_1), \\
\alpha_t(z_t, c_t) &= p(\mathbf{y}_{1:t}, \mathbf{x}_{1:t}, z_t, c_t) \\
&= \sum_{z_{t-1}}\sum_{c_{t-1}} p(\mathbf{y}_{1:t}, \mathbf{x}_{1:t}, z_{t-1}, z_t, c_{t-1}, c_t) \\
&= \sum_{z_{t-1}}\sum_{c_{t-1}} \Big[ p(\mathbf{y}_t, \mathbf{x}_t|\mathbf{x}_{t-1}, z_t)p(z_t|\mathbf{x}_{t-1}, z_{t-1}, c_t) \\
&\qquad\qquad\qquad \cdot p(c_t|z_{t-1}, c_{t-1})p(\mathbf{y}_{1:t-1}, \mathbf{x}_{1:t-1}, z_{t-1}, c_{t-1}) \Big] \\
&= \sum_{z_{t-1}}\sum_{c_{t-1}} p(\mathbf{y}_t, \mathbf{x}_t|\mathbf{x}_{t-1}, z_t)p(z_t|\mathbf{x}_{t-1}, z_{t-1}, c_t)p(c_t|z_{t-1}, c_{t-1})\alpha(z_{t-1}, c_{t-1}) \\
&= p(\mathbf{y}_t, \mathbf{x}_t|\mathbf{x}_{t-1}, z_t)\Bigg[ \delta_{\substack{z_{t-1}=z_t \\ c_t>1 \\ c_{t-1}=c_t-1}} v_{z_{t-1}}(c_{t-1})\alpha(z_{t-1}, c_{t-1}) \\
&\qquad + \delta_{c_t=1}\sum_{z_{t-1}} p(z_t|\mathbf{x}_{t-1}, z_{t-1}, c_t)\sum_{c_{t-1}}(1 - v_{z_{t-1}}(c_{t-1}))\alpha(z_{t-1}, c_{t-1}) \Bigg],
\end{aligned}
$$

$$
\begin{aligned}
\beta_T(z_T, c_T) &= 1, \\
\beta_t(z_t, c_t) &= p(\mathbf{y}_{t+1:T}, \mathbf{x}_{t+1:T}|\mathbf{x}_t, z_t, c_t) \\
&= \sum_{z_{t+1}, c_{t+1}} p(\mathbf{y}_{t+1:T}, \mathbf{x}_{t+1:T}, z_{t+1}, c_{t+1}|\mathbf{x}_t, z_t, c_t) \\
&= \sum_{z_{t+1}, c_{t+1}} \Big[ p(\mathbf{y}_{t+2:T}, \mathbf{x}_{t+2:T}|\mathbf{x}_{t+1}, z_{t+1}, c_{t+1})p(\mathbf{y}_{t+1}, \mathbf{x}_{t+1}|\mathbf{x}_t, z_{t+1}) \\
&\qquad\qquad\qquad \cdot p(z_{t+1}|\mathbf{x}_t, z_t, c_{t+1})p(c_{t+1}|z_t, c_t) \Big] \\
&= \delta_{\substack{z_{t+1}=z_t \\ c_{t+1}=c_t+1}} \beta_{t+1}(z_{t+1}, c_{t+1})p(\mathbf{y}_{t+1}, \mathbf{x}_{t+1}|\mathbf{x}_t, z_{t+1})v_{z_t}(c_t) \\
&\qquad + \delta_{c_t \geq d_{\min}}(1 - v_{z_t}(c_t))\sum_{\substack{c_{t+1}=1 \\ z_{t+1}}}\beta_{t+1}(z_{t+1}, c_{t+1})p(\mathbf{y}_{t+1}, \mathbf{x}_{t+1}|\mathbf{x}_t, z_{t+1})p(z_{t+1}|\mathbf{x}_t, z_t, c_{t+1}).
\end{aligned}
$$

## A.2 RED-SDS instantiation

In this section, we present our instantiation of RED-SDS, providing the general structure of the architecture and the functions $f_z$, $f_x^\mu$, $f_x^\Sigma$, $f_y^\mu$, $f_y^\Sigma$ (cf. Section 3.1). For specific implementation details we refer the reader to Appendices B.3 and C.3 for segmentation and forecasting, respectively.

**Control embedding network.** Control inputs can be either static, i.e., constant for all timesteps per time series (e.g., time series ID), or time-dependent (e.g., time embedding or other covariates). We denote the raw control input as $\mathbf{u}_{\text{raw}} = [\mathbf{u}^{\text{static}}, \mathbf{u}^{\text{time}}]$, where $\mathbf{u}^{\text{static}}$ and $\mathbf{u}^{\text{time}}$ correspond to the static and time-dependent control features, respectively.

In our forecasting experiments, we consider static features that are natural numbers (corresponding to time series IDs). The static features are first passed through an embedding layer. The embedding of the static features is then concatenated with the time-dependent features and the result is fed into a single hidden layer MLP that outputs the control input $\mathbf{u}_t \in \mathbb{R}^c$. This process is described by the following operations:

$$\mathbf{u}_t = f_u \left( g_{\text{emb}}^u(\mathbf{u}_t^{\text{static}}), \mathbf{u}_t^{\text{time}} \right), \tag{20}$$

where $g_{\text{emb}}^u$ denotes the embedding layer and $f_u$ is the MLP.

**Inference network.** The observations $\mathbf{y}_{1:T}$ are first passed through an embedding network $g_{\text{emb}}^y$, that can either be a bi-RNN or a Transformer, which outputs an embedding $\mathbf{h}_{1:T}^1$ of the observations. The embedding is then fed into a causal RNN $g_{\text{rnn}}$ along with the control inputs $\mathbf{u}_{1:T}$ which outputs hidden states $\mathbf{r}_{1:T}$. A single hidden layer MLP $g_{\text{fc}}$ then maps the hidden states $\mathbf{r}_{1:T}$ to the parameters of the Gaussian distribution $q(\mathbf{x}_t|\mathbf{x}_{1:t-1}, \mathbf{h}_{1:T}^1)$. This inference network is described by the following operations:

$$\mathbf{h}_{1:T}^1 = g_{\text{emb}}^y(\mathbf{y}_{1:T}), \tag{21}$$

$$\mathbf{r}_t = g_{\text{rnn}}(\mathbf{x}_{t-1}, \mathbf{r}_{t-1}, \mathbf{u}_t, \mathbf{h}_t^1), \tag{22}$$

$$\mathbf{x}_t \sim \mathcal{N} \left( \mathbf{x}_t; g_{\text{fc}}^\mu(\mathbf{r}_t), g_{\text{fc}}^\Sigma(\mathbf{r}_t) \right), \tag{23}$$

where $\mathbf{x}_0$ and $\mathbf{r}_0$ are zero vectors.

**Duration network.** When no control input is available, the duration model is a learnable matrix $\mathbf{P} \in \mathbb{R}^{K \times (d_{\max} - d_{\min} + 1)}$, fixed across all timesteps; the $k$-th row represents the logits of the duration model $\rho_k$ for the $k$-th switch. When control inputs are used, the duration model depends on the control input and is no longer fixed across timesteps. The control input $\mathbf{u}_t$ (see Eq. 20) is fed into a single hidden layer MLP which outputs a (time-varying) matrix $\mathbf{P}_t \in \mathbb{R}^{K \times d_{\max}}$, i.e.,

$$\mathbf{P}_t = f_d(\mathbf{u}_t). \tag{24}$$

where $\mathbf{P}_t$ has the same structure as $\mathbf{P}$. The first $d_{\min} - 1$ columns are masked and the final duration models $\rho_k$ are obtained by applying the tempered softmax function to the rows of $\mathbf{P}_t$,

$$\rho_k(d)_t = \text{Cat}(d; \mathcal{S}_{\tau_\rho}(\mathbf{P}_t^{k,:})), \tag{25}$$

where $\mathbf{P}_t^{k,:}$ denotes the $k$-th row of $\mathbf{P}_t$ and $\mathcal{S}_{\tau_\rho}$ denotes the tempered softmax function with temperature $\tau_\rho$.

**Discrete transition network.** The pseudo-observations of the states $\mathbf{x}_1, \ldots, \mathbf{x}_{t-1}$, sampled from the inference network, along with the control inputs $\mathbf{u}_2, \ldots, \mathbf{u}_T$, are passed to the neural network $f_z$ (for the case when $c_t = 1$), a single hidden layer MLP, to model the discrete transition distribution,

$$p(z_t|\mathbf{x}_{t-1}, z_{t-1}, c_t, \mathbf{u}_t) = \begin{cases} \delta_{z_t = z_{t-1}} & \text{if } c_t > 1 \\ \text{Cat}(z_t; \mathcal{S}_{\tau_z}(f_z(\mathbf{x}_{t-1}, z_{t-1}, \mathbf{u}_t))) & \text{if } c_t = 1 \end{cases}, \tag{26}$$

where $\mathcal{S}_{\tau_z}$ denotes the tempered softmax function with temperature $\tau_z$. The network $f_z$ takes $\mathbf{x}_{t-1}$ and $\mathbf{u}_t$ as input and outputs a matrix $\tilde{\mathbf{A}}_t \in \mathbb{R}^{K \times K}$. Each row of the matrix $\tilde{\mathbf{A}}_t \in \mathbb{R}^{K \times K}$ is normalized using $\mathcal{S}_{\tau_z}$ to obtain the stochastic transition matrix $\mathbf{A}_t$ where each row represents a categorical distribution that can be indexed by $z_{t-1}$.

**Continuous transition network.** The continuous transition network $f_x$ is a linear function or a single hidden layer MLP that models the continuous transition distribution

$$p(\mathbf{x}_t|\mathbf{x}_{t-1}, z_t, \mathbf{u}_t) = \mathcal{N}\left(\mathbf{x}_t; f_x^\mu(\mathbf{x}_{t-1}, z_t, \mathbf{u}_t), f_x^\Sigma(\mathbf{x}_{t-1}, z_t, \mathbf{u}_t)\right). \tag{27}$$

The function $f_x$ takes $\mathbf{x}_{t-1}$ and $\mathbf{u}_t$ as input and outputs the parameters of the Gaussian distribution. The dependence on $z_t$ is realized by using separate functions $f_x^k$ for the $K$ unique values of the switch $z_t$.

**Emission network.** The emission network $f_y$ is a linear function or an MLP with two hidden layers that models the emission distribution

$$p(\mathbf{y}_t|\mathbf{x}_t) = \mathcal{N}\left(\mathbf{y}_t; f_y^\mu(\mathbf{x}_t), f_y^\Sigma(\mathbf{x}_t)\right). \tag{28}$$

### A.3 Applications

In this section, we describe how to perform time series segmentation and generate probabilistic forecasts using RED-SDS.

#### A.3.1 Segmentation

We perform time series segmentation by labeling every timestep with the most likely switch. This is done by first computing the posterior distribution $\gamma_t(z_t, c_t)$ (Eq. 19) for each timestep and then obtaining the most likely switch $\hat{k}_t$ by marginalizing out the count variable $c_t$ as follows

$$\hat{k}_t = \underset{j}{\arg\max} \sum_{d=d_{\min}}^{d_{\max}} \gamma_t(z_t = j, c_t = d). \tag{29}$$

#### A.3.2 Forecasting

We generate probabilistic forecasts by generating multiple future sample paths. Let $\mathbf{y}_{1:T}$ be an input time series and $\tau$ the forecast horizon. We begin by generating $M$ state samples from the variational posterior $q(\mathbf{x}_T|\mathbf{y}_{1:T})$ and the corresponding $M$ switch-count pairs from the posterior over the switches and counts $p(z_T, c_T|\mathbf{y}_{1:T}, \hat{\mathbf{x}}_{1:T})$. These $M$ triplets of state, switch, and count are then used to unroll the generative model into the future for $\tau$ timesteps generating $M$ sample paths (forecasts). Algorithm 1 describes the steps involved to generate one such sample path.

---

**Algorithm 1** RED-SDS future unrolling

INPUT
    Time series $\mathbf{y}_{1:T}$, forecast horizon $\tau$

SAMPLE STATES AT $T$
    Continuous state sample from the variational posterior $\hat{\mathbf{x}}_T \sim q(\mathbf{x}_T|\mathbf{y}_{1:T})$
    Discrete state and duration samples $\hat{z}_T, \hat{c}_T \sim p(z_T, c_T|\mathbf{y}_{1:T}, \hat{\mathbf{x}}_{1:T}) = \gamma_T(z_T, c_T)$, where $\gamma_T(z_T, c_T)$ is computed using the forward-backward algorithm
    Set $\hat{\mathbf{y}}_T = \mathbf{y}_T$

UNROLL IN FORECAST HORIZON
    **for** $t = T + 1 : T + \tau$ **do**
        $\hat{c}_t \sim p(c_t|\hat{z}_{t-1}, \hat{c}_{t-1})$
        $\hat{z}_t \sim p(z_t|\hat{\mathbf{x}}_{t-1}, \hat{z}_{t-1}, \hat{c}_t)$
        $\hat{\mathbf{x}}_t \sim p(\mathbf{x}_t|\hat{\mathbf{x}}_{t-1}, \hat{z}_t)$
        $\hat{\mathbf{y}}_t \sim p(\mathbf{y}_t|\hat{\mathbf{x}}_t)$
    **end for**

RETURN
    Predictive samples $\hat{\mathbf{y}}_{T+1:T+\tau}$

---

# B  Details on segmentation experiments

## B.1  Datasets

**Bouncing ball.**  The bouncing ball dataset comprises univariate time series $y_{1:T}$, with $y_t \in \mathbb{R}$, that encode the location of a ball bouncing between two fixed walls with constant absolute velocity and elastic collisions between the ball and the wall(s). The distance between the walls was set to 10 with the initial location of the ball randomly generated between the walls. The initial velocity was sampled from the uniform distribution $\mathcal{U}(-0.5, 0.5)$ and its sign was flipped every time the ball went beyond 0 or 10. The final observation was generated by adding $\epsilon \sim \mathcal{N}(0, 0.1^2)$ to the ball's location. The ground truth label was assigned based on the sign of the velocity. We generated 100000 and 1000 time series of 100 timesteps for the train and the test datasets respectively.

**3 mode system.**  The 3 mode system dataset comprises univariate time series generated from a switching linear dynamical system with 3 modes and an explicit duration model for each mode. The dimensionality of the state variables $\mathbf{x}_t$ was set to 2. The initial switch and state distributions were given by

$$p(z_1) = \mathcal{U}_{\text{cat}}\{1, 2, 3\}, \tag{30}$$

$$p(\mathbf{x}_1|z_1) = \mathcal{N}\left(\begin{bmatrix} 2 \\ 0 \end{bmatrix}, 0.01\mathbf{I}\right), \tag{31}$$

where $\mathcal{U}_{\text{cat}}$ denotes the uniform categorical distribution.

The transition distribution of the increasing count variables was given by

$$p(c_t|z_{t-1} = k, c_{t-1}) = \begin{cases} v_k(c_{t-1}) & \text{if} \quad c_t = c_{t-1} + 1 \\ 1 - v_k(c_{t-1}) & \text{if} \quad c_t = 1 \end{cases}, \tag{32}$$

where the probability of a count increment $v_k$ for the switch $k$ is defined as in Eq. (2) via the duration models $\rho_k$, given by

$$\rho_1(d) = \text{Cat}\left(\begin{bmatrix} \frac{2}{17} & 0 & 0 & 0 & 0 & \frac{5}{17} & 0 & 0 & 0 & 0 & \frac{7}{17} & 0 & 0 & 0 & \frac{3}{17} \end{bmatrix}\right), \tag{33}$$

$$\rho_2(d) = \text{Cat}\left(\begin{bmatrix} 0 & 0 & \frac{1}{4} & 0 & 0 & 0 & 0 & 0 & 0 & 0 & 0 & \frac{2}{5} & 0 & \frac{3}{10} & \frac{1}{20} \end{bmatrix}\right), \tag{34}$$

$$\rho_3(d) = \text{Cat}\left(\begin{bmatrix} 0 & 0 & 0 & 0 & 0 & 0 & 0 & \frac{3}{17} & 0 & 0 & \frac{7}{17} & 0 & \frac{5}{17} & 0 & \frac{2}{17} \end{bmatrix}\right), \tag{35}$$

with $d_{\min} = 6$ and $d_{\max} = 20$.

The switch transition distribution was given by

$$p(z_t = j|z_{t-1} = i, c_t) = \begin{cases} \delta_{z_t = i} & \text{if} \quad c_t > 1 \\ \begin{bmatrix} 0.1 & 0.2 & 0.7 \\ 0.3 & 0.5 & 0.2 \\ 0.3 & 0.3 & 0.4 \end{bmatrix}_{(i,j)} & \text{if} \quad c_t = 1 \end{cases}. \tag{36}$$

The state transition distribution was given by

$$p(\mathbf{x}_t|\mathbf{x}_{t-1}, z_t = k) = \mathcal{N}(\mathbf{A}_k\mathbf{x}_{t-1} + \mathbf{b}_k, 0.01\mathbf{I}), \tag{37}$$

where the $\mathbf{A}$s and $\mathbf{b}$s were defined as follows

$$\mathbf{A}_1 = 0.99\begin{bmatrix} \cos(0) & -\sin(0) \\ \sin(0) & \cos(0) \end{bmatrix}, \tag{38}$$

$$\mathbf{b}_1 = \mathbf{0}, \tag{39}$$

$$\mathbf{A}_2 = 0.99\begin{bmatrix} \cos(\pi/8) & -\sin(\pi/8) \\ \sin(\pi/8) & \cos(\pi/8) \end{bmatrix}, \tag{40}$$

$$\mathbf{b}_2 = 0.25\epsilon_2, \tag{41}$$

$$\mathbf{A}_3 = 0.99\begin{bmatrix} \cos(\pi/4) & -\sin(\pi/4) \\ \sin(\pi/4) & \cos(\pi/4) \end{bmatrix}, \tag{42}$$

$$\mathbf{b}_3 = 0.25\epsilon_3, \tag{43}$$

with $\epsilon_2, \epsilon_3$ sampled from $\mathcal{N}(\mathbf{0}, \mathbf{I})$.

The emission distribution was given by

$$p(\mathbf{y}_t|\mathbf{x}_t, z_t = k) = \mathcal{N}\left(\mathbf{c}_k^\top \mathbf{x}_t + d_k, 0.04\mathbf{I}\right), \tag{44}$$

with $\mathbf{c}_k \sim \mathcal{N}(\mathbf{0}, \mathbf{I})$ and $d_k \sim \mathcal{U}_{\text{cat}}\{0, 1, 2\}$.

The ground truth label was assigned based on $z_t$. We generated 10000 and 500 time series of 180 timesteps for the train and the test datasets, respectively.

**Dancing bees.** We used the publicly-available[3] dancing bees dataset [40], which comprises tracks of six dancer honey bees that were obtained using a vision-based tracker. The time series consist of the 2D coordinates $(x_t, y_t)$ and the heading angle $(\theta_t)$ of the honey bee at every timestep. Each timestep is labeled as one of the three dance types: waggle, turn right, and turn left. The six long time series have lengths 1058, 1125, 1054, 757, 609 and 814. We first standardized the 2D coordinates $(x_t, y_t)$ for each time series by subtracting the mean and dividing by the standard deviation to get the normalized coordinates $(\hat{x}_t, \hat{y}_t)$. We then constructed 4-dimensional time series with elements $[\hat{x}_t, \hat{y}_t, \sin(\theta_t), \cos(\theta_t)]$. Each time series was split into chunks of 120 timesteps starting at every changepoint in the ground truth label (discarding terminal chunks less than 120 timesteps). We used time series $\{1, 3, 4, 5, 6\}$ for training and the longest time series $\{2\}$ for testing, resulting in 84 and 21 data points in the train and test datasets, respectively.

## B.2 Segmentation metrics

The segment label ordering assigned by a model such as RED-SDS—trained without label supervision—is arbitrary and may not match with the ground truth label ordering. Thus, we evaluated the segmentation performance using three metrics that are independent of remappings of the predicted labels:

(a) Frame-wise segmentation accuracy: This metric computes the percentage of predicted labels that match the ground truth labels. The predicted labels were first remapped using the Hungarian algorithm [29] which finds the mapping that maximizes the accuracy. We used `linear_sum_assignment` from `scipy` to find the optimal mapping.

(b) Normalized Mutual Information (NMI) [47]: NMI computes the mutual information between two labelings, normalized to lie between 0 (no mutual information) and 1 (perfect correlation). The metric is independent of the actual values of labels, i.e., a remapping of the labels won't change the score. We used `normalized_mutual_info_score` from `scikit-learn` to compute the NMI score.

(c) Adjusted Rand Index (ARI) [23]: ARI computes a similarity measure between two labelings which is adjusted for chance. The metric lies between -1 to 1 where a random labeling has a score close to 0 and a score of 1 indicates a perfect match. Like NMI, ARI is independent of the actual values of the labels. We used `adjusted_rand_score` from `scikit-learn` to compute the ARI score.

## B.3 Training and hyperparameter details

In this section, we discuss the training and hyperparameter details for the segmentation experiments.

**Training parameters.** We trained all the datasets with a fixed batch size of 32 for 20000 training steps. We used the Adam optimizer for the gradient updates with $10^{-5}$ weight-decay and clipped the gradient norm to 10. The learning rate was warmed-up linearly from a lower value in $\{5 \times 10^{-5}, 1 \times 10^{-4}\}$ to $\{2 \times 10^{-4}, 5 \times 10^{-3}\}$ for the first $\{2000, 1000\}$ steps after which a cosine decay follows for the remaining time steps with a decay rate of 0.99.

**Network types.** We experimented with linear and non-linear functions for the continuous transition $f_x$ and emission $f_y$ functions. The linear function implies multiplication with a transformation matrix while the non-linear function was an MLP with ReLU non-linearity. For the inference embedding

---

[3]`https://www.cc.gatech.edu/~borg/ijcv_psslds/`

Table 3: Network architectures for the different components of the model. `Linear` denotes a linear layer without bias, `MLP` $[a_1, \ldots, a_l]$ denotes an $l$-hidden-layer MLP with hidden units $a_1, \ldots, a_l$ and ReLU non-linearity, `biGRU` $[b]$ denotes a single-layer bidirectional GRU with $b$ hidden units, and `RNN` $[c]$ denotes a single-layer RNN with $c$ hidden units.

| Network | Datasets | | |
| --- | --- | --- | --- |
| | `bouncing ball` | `3 mode system` | `dancing bees` |
| Discrete Transition ($f_z$) | MLP $[4 \times 2^2]$ | MLP $[4 \times 3^2]$ | MLP $[4 \times 3^2]$ |
| Continuous Transition ($f_x$) | MLP $[32]$ | MLP $[32]$ | Linear |
| Emission Network ($f_y$) | MLP $[8, 32]$ | MLP $[8, 32]$ | Linear |
| Inference Embedder ($g_{\text{emb}}^y$) | biGRU $[4]$ | biGRU $[4]$ | biGRU $[16]$ |
| Causal RNN ($g_{\text{rnn}}$) | RNN $[16]$ | RNN $[16]$ | RNN $[16]$ |
| Parameter Network ($g_{\text{fc}}$) | MLP $[32]$ | MLP $[32]$ | MLP $[32]$ |

Table 4: Temperature annealing details.

(a) Annealing schedules of the switch ($\tau_z$) and duration ($\tau_\rho$) temperatures.

| Hyperparameter | Datasets | | |
| --- | --- | --- | --- |
| | `bouncing ball` | `3 mode system` | `dancing bees` |
| Initial Temp. | $\tau_z, \tau_\rho$: 10 | $\tau_z, \tau_\rho$: 10 | $\tau_z$: 100, $\tau_\rho$: 10 |
| Min Temp. | $\tau_z, \tau_\rho$: 1 | $\tau_z, \tau_\rho$: 1 | $\tau_z$: 10, $\tau_\rho$: 1 |
| Decay Rate | 0.99 | 0.99 | 0.95 |
| Begin Decay Step | 1000 | 1000 | 5000 |
| Decay Every (steps) | 50 | 50 | 100 |

(b) ✓ indicates that the temperature is annealed using the schedule in (a) and ✗ indicates that the temperature is kept fixed.

| Model | Variable | Datasets | | |
| --- | --- | --- | --- | --- |
| | | `bouncing ball` | `3 mode system` | `dancing bees` |
| SNLDS | Switch | ✓ | ✓ | ✓ |
| ED-SDS | Switch | ✗ | ✗ | ✓ |
| | Duration | ✓ | ✓ | ✓ |
| RED-SDS | Switch | ✗ | ✗ | ✓ |
| | Duration | ✗ | ✓ | ✓ |

network $g_{\text{emb}}^y$, we used a single layer bidirectional GRU. The causal RNN $g_{\text{rnn}}$ was a single layer forward RNN and the parameter network $g_{\text{fc}}$ was a single hidden layer MLP. Table 3 summarizes the network architectures of the different model components for the three datasets.

**Temperature Annealing.**    As discussed in Section 3.3, we annealed the temperature parameter of the tempered softmax function (Eq. 16) to encourage the model to explore all switches and durations during the initial training phase. We decayed the temperature parameter exponentially from a initial value to a minimum value and report the details on annealing schedules in Table 4.

**Duration.**    We used the $(d_{\min}, d_{\max})$ hyperparameters to impart prior knowledge to the model about the switch durations. These parameters were set to $(1, 20)$ for the bouncing ball dataset. For the 3 mode system and dancing bees datasets, we used larger values of $(5, 20)$ and $(20, 50)$ respectively to encourage the model to learn longer switch durations as exhibited by these datasets.

**Number of switches.**    The number of switches $K$ plays an important role in the model, particularly for segmentation. For all the datasets considered in our experiments, the number of operating modes is known; therefore, we set $K$ to the number of ground truth operating modes.

**Dimensionality of state $\mathbf{x}_t$.** The dimensionality of the state variables $\mathbf{x}_t$ was tuned from the set $\{2, 4\}$ for the univariate datasets (bouncing ball and 3 mode system) and was set to 8 for the multivariate dataset (dancing bees).

### B.4 Baseline models

#### B.4.1 SNLDS

In this section, we discuss the objective function of SNLDS proposed by Dong et al. [13] and the modified version used in our experiments. As mentioned in the main text, SNLDS can be viewed as an ablated version of RED-SDS without the explicit duration models, i.e., with $d_{\min} = d_{\max} = 1$.

The ELBO for SNLDS is given by

$$\text{ELBO} = \mathbb{E}_{q(\mathbf{x}_{1:T}|\mathbf{y}_{1:T})p(z_{1:T}|\mathbf{y}_{1:T},\mathbf{x}_{1:T})} \left[ \log \frac{p(\mathbf{y}_{1:T}, \mathbf{x}_{1:T}, z_{1:T})}{q(\mathbf{x}_{1:T}|\mathbf{y}_{1:T})p(z_{1:T}|\mathbf{y}_{1:T},\mathbf{x}_{1:T})} \right] \tag{45}$$

$$= \mathbb{E}_{q(\mathbf{x}_{1:T}|\mathbf{y}_{1:T})} \left[ \log \frac{p(\mathbf{y}_{1:T}, \mathbf{x}_{1:T})}{q(\mathbf{x}_{1:T}|\mathbf{y}_{1:T})} \right]. \tag{46}$$

To estimate the gradients of $\log p(\mathbf{y}_{1:T}, \mathbf{x}_{1:T})$ in Eq. (46), Dong et al. [13] proposed auto-differentiating through the following expression,

$$\mathbb{E}_{p(z_{1:T}|\mathbf{y}_{1:T},\mathbf{x}_{1:T})} \left[ \log p(\mathbf{y}_{1:T}, \mathbf{x}_{1:T}, z_{1:T}) \right] = \sum_{t=2}^{T} \sum_{j,k} \xi_t(j,k) \left[ \log B_t(k) A_t(j,k) \right]$$
$$+ \sum_k \gamma_1(k) \left[ \log B_1(k) \pi_k \right], \tag{47}$$

where

$$\pi(k) = p(z_1 = k), \tag{48}$$
$$\gamma(k) = p(z_t = k|\mathbf{y}_{1:T}, \mathbf{x}_{1:T}), \tag{49}$$
$$\xi(j,k) = p(z_t = k, z_{t-1} = j|\mathbf{y}_{1:T}, \mathbf{x}_{1:T}), \tag{50}$$
$$B_t(k) = p(\mathbf{y}_t|\mathbf{x}_t)p(\mathbf{x}_t|\mathbf{x}_{t-1}, z_t = k), \tag{51}$$
$$A_t(j,k) = p(z_t = j|\mathbf{y}_{t-1}, z_{t-1} = k). \tag{52}$$

However, this makes the model prone to state collapse, i.e., the model ends up only using a single switch, as also noted by [13]. This led them to add an ad-hoc cross-entropy regularizer to the objective function that discourages state collapse during the initial phase of training. The regularizer minimizes the KL divergence between a uniform prior on the switches and the smoothed posterior over switches,

$$\mathcal{L}_{\text{CE}} = \sum_{t=1}^{T} \mathcal{D}_{KL}(p_{\text{prior}}(z_t)\|\gamma(z_t)), \tag{53}$$

where $p_{\text{prior}}(z_t) = \prod_j \left[ \frac{1}{K} \right]^{\mathbf{1}[j=z_t]}$.

During our initial experiments with SNLDS, we observed frequent state collapse when using the approximation in Eq. (47) and the training was sensitive to the annealing schedule of $\mathcal{L}_{\text{CE}}$. We further noted that the likelihood term $p(\mathbf{y}_{1:T}, \mathbf{x}_{1:T})$ in Eq. (46) can be computed exactly (same as in RED-SDS) using the SNLDS forward variable $\alpha_T(z_T) = p(\mathbf{y}_{1:T}, \mathbf{x}_{1:T}, z_T)$ as

$$p(\mathbf{y}_{1:T}, \mathbf{x}_{1:T}) = \sum_{z_T} \alpha_T(z_T). \tag{54}$$

Therefore, we used Eq. (54) for $p(\mathbf{y}_{1:T}, \mathbf{x}_{1:T})$ in our experiments which was less prone to state collapse and didn't require the sensitive KL regularization term (Eq. 53).

#### B.4.2 Soft-switching models

We conducted experiments using two soft-switching models: ARSGLS and KVAE. The ARSGLS extends the classical SLDS by conditionally linear state-to-switch connections and an auxiliary

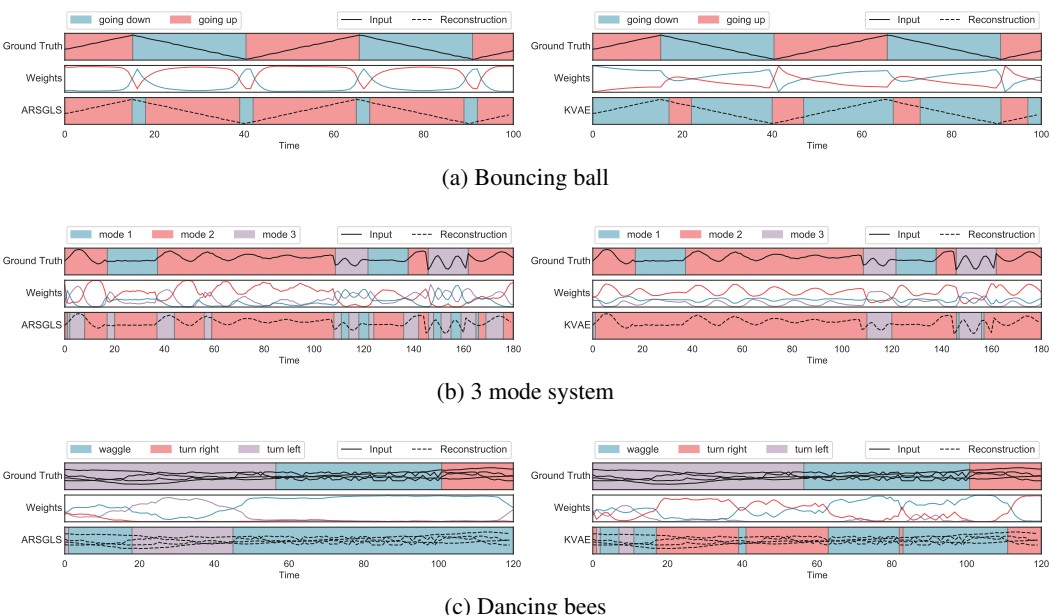

(a) Bouncing ball

(b) 3 mode system

(c) Dancing bees

Figure 6: Qualitative segmentation results on the bouncing ball, 3 mode system, and dancing bees datasets for the soft-switching models: ARSGLS (left) and KVAE (right). Background colors represent the different operating modes.

variable along with a neural network (decoder) that allows it to model more complex non-linear emission models and multivariate observations. However, the switch variables in ARSGLS are Gaussian rather than categorical, such that the recurrent state-to-switch connection does not break the computational tractability of the conditionally linear dynamical system. Instead of "indexing" a distinct base LDS at each timestep, the parameters are predicted as a weighted average of the matrices of the base LDSs. These combination weights are predicted by a neural network with a softmax transformation which takes the Gaussian switch variables as input.

Similarly, the KVAE uses an LSTM (followed by an MLP) to predict the weights for averaging the base matrices, where the LSTM takes the latent variables embedded by a variational autoencoder as inputs.

As both ARSGLS and KVAE are trained using a continuous interpolation of operating modes, they cannot always correctly assign a single discrete operating mode to every timestep during test time. Fig. 6 shows qualitative segmentation results for ARSGLS and KVAE trained on the bouncing ball, 3 mode system, and dancing bees datasets along with the combination weights at each timestep (middle of each plot). The segment labels were assigned by taking the $\arg\max$ of the combination weights. Although the combination weights follow some interpretable patterns (e.g., in the bouncing ball and 3 mode system datasets), they cannot assign each timestep to a single mode. Thus, both ARSGLS and KVAE perform poorly on these segmentation tasks.

## B.5 Additional results

Table 5 presents segmentation results on the bouncing ball and 3 mode system datasets with the dimenionality of the state $\mathbf{x}_t$ equal to 2 (ground truth) and 4. All the three models perform similarly for $\dim(\mathbf{x}_t) = 2$ and $\dim(\mathbf{x}_t) = 4$ on the bouncing ball dataset. On the 3 mode system dataset, RED-SDS performs better with $\dim(\mathbf{x}_t) = 4$ than $\dim(\mathbf{x}_t) = 2$.

Fig. 7 shows the duration models learned by ED-SDS and RED-SDS. On the bouncing ball dataset, RED-SDS assigns all the probability mass to shorter durations indicating that the state-to-switch recurrence is more informative about the switching behavior in this dataset. In contrast, ED-SDS lacks state-to-switch recurrence and assigns probability mass to longer durations. On the 3 mode system dataset, both ED-SDS and RED-SDS recover the ground truth duration model. On the dancing

bees dataset, both ED-SDS and RED-SDS assign probability mass to longer durations indicating the existence of long-term temporal patterns.

Table 5: Quantitative results on the bouncing ball and 3 mode system datasets with different values of $\dim(\mathbf{x}_t)$.

| Metric | Model | $\dim(\mathbf{x}_t)$ | Datasets | |
| --- | --- | --- | --- | --- |
| | | | bouncing ball | 3 mode system |
| Accuracy | SNLDS | 2 | **0.97±0.00** | 0.82±0.14 |
| | | 4 | **0.97±0.00** | 0.82±0.08 |
| | ED-SDS (ours) | 2 | 0.95±0.00 | 0.97±0.00 |
| | | 4 | 0.94±0.00 | 0.97±0.00 |
| | RED-SDS (ours) | 2 | **0.97±0.00** | 0.95±0.00 |
| | | 4 | **0.97±0.00** | **0.98±0.00** |
| NMI | SNLDS | 2 | **0.83±0.01** | 0.63±0.12 |
| | | 4 | 0.82±0.01 | 0.63±0.08 |
| | ED-SDS (ours) | 2 | 0.71±0.00 | 0.89±0.00 |
| | | 4 | 0.70±0.01 | 0.88±0.01 |
| | RED-SDS (ours) | 2 | 0.80±0.00 | 0.82±0.01 |
| | | 4 | 0.81±0.00 | **0.91±0.01** |
| ARI | SNLDS | 2 | **0.90±0.01** | 0.67±0.17 |
| | | 4 | 0.89±0.01 | 0.67±0.11 |
| | ED-SDS (ours) | 2 | 0.81±0.01 | 0.93±0.00 |
| | | 4 | 0.79±0.01 | 0.93±0.01 |
| | RED-SDS (ours) | 2 | 0.88±0.01 | 0.89±0.01 |
| | | 4 | 0.88±0.00 | **0.95±0.01** |

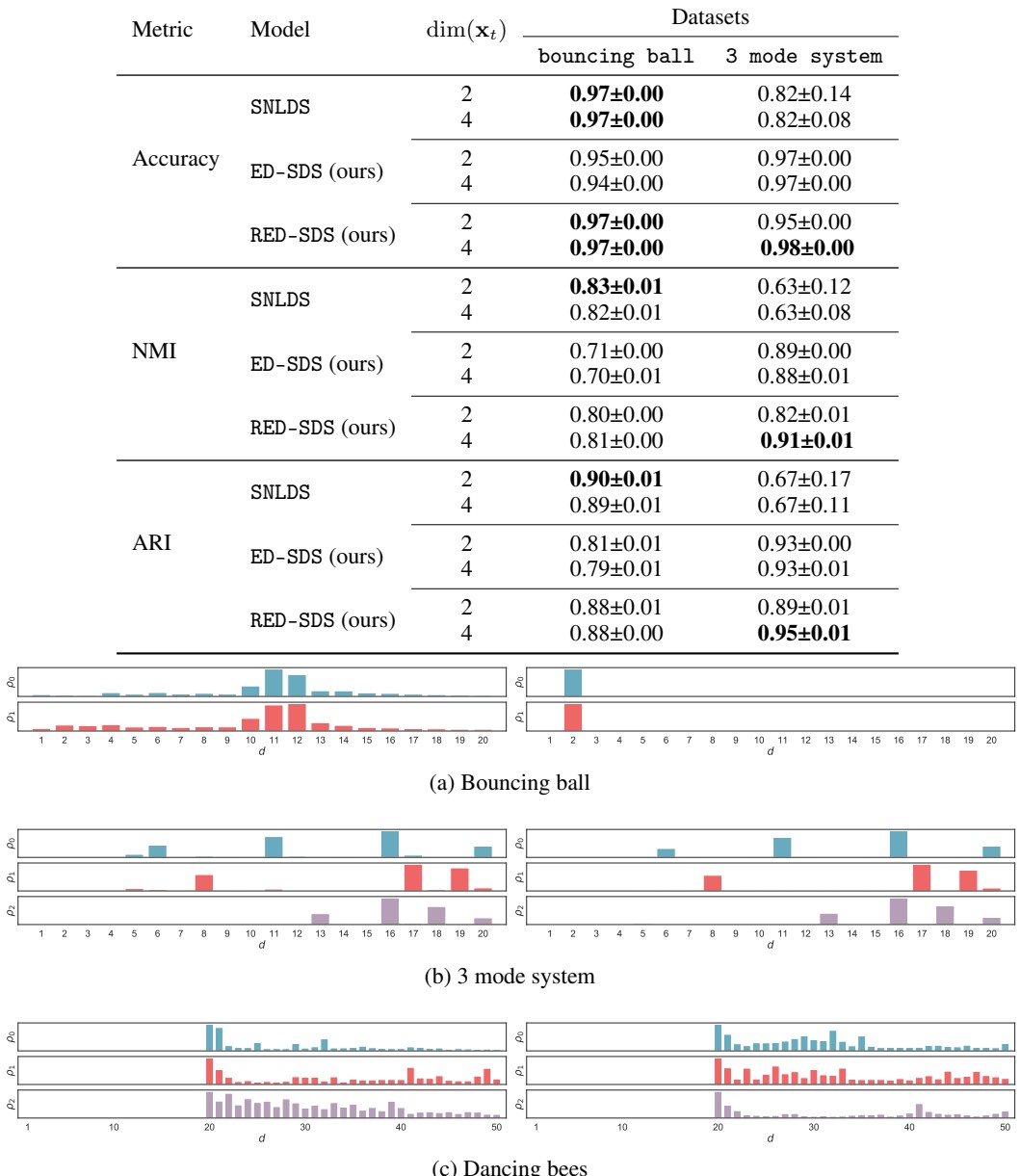

(a) Bouncing ball

(b) 3 mode system

(c) Dancing bees

Figure 7: Duration models learned by ED-SDS (left) and RED-SDS (right) for the bouncing ball, 3 mode system, and dancing bees datasets.

## C    Details on forecasting experiments

### C.1    Datasets

We evaluated RED-SDS on the following five publicly available datasets, commonly used for evaluation forecasting models:

- `exchange`: daily exchange rate between 8 currencies as used in [31];

- `solar`: hourly photo-voltaic production of 137 stations in Alabama State used in [31];
- `electricity`: hourly time series of the electricity consumption of 370 customers [15];
- `traffic`: hourly occupancy rate, between 0 and 1, of 963 San Francisco car lanes [15];
- `wiki`: daily page views of 2000 Wikipedia pages used in [20].

Similar to [30], the forecasts of different methods are evaluated by splitting each dataset in the following fashion: all data prior to a fixed *forecast start date* comprise the training set and the remainder is used as the test set.

In Table 6, we provide an overview of the different properties of these datasets.

Table 6: Overview of the datasets used to test the forecasting accuracy of the models.

| dataset | domain | frequency | # of time series | # of timesteps | $T$ (context length) | $\tau$ (forecast horizon) |
|---|---|---|---|---|---|---|
| exchange | $\mathbb{R}^+$ | daily | 8 | 6071 | 124 | 150 |
| solar | $\mathbb{R}^+$ | hourly | 137 | 7009 | 336 | 168 |
| electricity | $\mathbb{R}^+$ | hourly | 370 | 5790 | 336 | 168 |
| traffic | $\mathbb{R}^+$ | hourly | 963 | 10413 | 336 | 168 |
| wiki | $\mathbb{N}$ | daily | 2000 | 792 | 124 | 150 |

## C.2  Forecasting metric: CRPS

The continuous ranked probability score (CRPS) [37] is a proper scoring rule [22] that measures the compatibility of a quantile function $F^{-1}$ with an observation $y$. It has an intuitive definition as the pinball loss integrated over all quantile levels $\alpha \in [0, 1]$, i.e.,

$$\text{CRPS}(F^{-1}, y) = \int_0^1 2\Lambda_\alpha(F^{-1}(\alpha), y)\, d\alpha, \tag{55}$$

where the pinball loss $\Lambda_\alpha(q, y)$ is defined as

$$\Lambda_\alpha(q, y) = (\alpha - \mathbb{1}_{\{y < q\}})(y - q), \tag{56}$$

with $q$ being the respective quantile of the probability distribution.

We used the CRPS implementation provided in GluonTS [4] that approximates the integral of (55) with a grid of selected quantiles.

## C.3  Training and hyperparameter details

In this section we provide the training and hyperparameter details for the forecasting experiments.

**Training parameters.**  We trained all the datasets with a fixed batch size of 50 and tuned the number of training steps per dataset. We used the Adam optimizer for the gradient updates with $10^{-5}$ weight-decay and clipped the gradient norm to 10. The learning rate was warmed-up linearly from $1 \times 10^{-4}$ to a higher value in the range $[5 \times 10^{-4}, 1 \times 10^{-2}]$ (optimized per dataset) for the first 1000 steps after which a cosine decay follows for the remaining time steps with a decay rate of 0.99.

**Network types.**  We experimented with linear and non-linear functions for the continuous transition $f_x$ and emission $f_y$ functions as previously described in Appendix B.3. For the inference embedding network $g_{\text{emb}}^y$, we used a single transformer layer with one attention head. The input time series was first mapped into a 4-dimensional embedding and then concatenated with the positional encoding before feeding into the transformer layer. For the control network ($f_u$) and the duration network ($f_d$) we used single hidden layer MLPs. Table 7 summarizes the network architectures of the different model components for the five forecasting datasets.

**Control embedding.**  As described in subsection A.2, the raw control features are comprised of static features $\mathbf{u}^{\text{static}}$ and time features $\mathbf{u}^{\text{time}}$. In our experiments, the static features are the time series IDs. Time series ID are first fed into an embedding layer that outputs a $p$-dimensional embedding; this time series embedding is concatenated with the time features and passed to the control network ($f_u$) to output a $c$-dimensional control $\mathbf{u}_t$. Table 8 lists the values of $p$ and $c$ for the different datasets.

Table 7: Network architectures for the different components of RED-SDS for forecasting experiments. `Linear` denotes a linear layer without bias, `MLP` $[a_1, \ldots, a_l]$ denotes a $l$-hidden-layer MLP with hidden units $a_1, \ldots, a_l$ and ReLU non-linearity, `biGRU` $[b]$ denotes a single-layer bidirectional GRU with $b$ hidden units, `RNN` $[c]$ denotes a single-layer RNN with $c$ hidden units, and `Transformer` denotes a transformer layer with one attention head and the dimension of feedforward network equal to 16.

| Network | Datasets | | | | |
| --- | --- | --- | --- | --- | --- |
| | `exchange` | `solar` | `electricity` | `traffic` | `wiki` |
| Control Network ($f_u$) | MLP [32] | MLP [32] | MLP [32] | MLP [64] | MLP [32] |
| Duration Network ($f_d$) | MLP [64] | MLP [64] | MLP [64] | MLP [64] | MLP [64] |
| Discrete Transition ($f_z$) | MLP $[4 \times 2^2]$ | MLP $[4 \times 4^2]$ | MLP $[4 \times 3^2]$ | MLP $[4 \times 4^2]$ | MLP $[4 \times 5^2]$ |
| Continuous Transition ($f_x$) | MLP [32] | Linear | MLP [32] | MLP [32] | Linear |
| Emission Network ($f_y$) | MLP [8, 32] | Linear | MLP [8, 32] | MLP [8, 32] | Linear |
| Inference Embedder ($g^y_{\text{emb}}$) | biGRU [4] | Transformer | Transformer | Transformer | Transformer |
| Causal RNN ($g_{\text{rnn}}$) | RNN [16] | RNN [16] | RNN [16] | RNN [16] | RNN [16] |
| Parameter Network ($g_{\text{fc}}$) | MLP [32] | MLP [32] | MLP [32] | MLP [32] | MLP [32] |

Table 8: Control embedding hyperparameters.

| Hyperparameter | Datasets | | | | |
| --- | --- | --- | --- | --- | --- |
| | exchange | solar | electricity | traffic | wiki |
| $p$ | 5 | 8 | 32 | 50 | 8 |
| $c$ | 16 | 16 | 32 | 32 | 32 |

**Duration.** We set $(d_{\min}, d_{\max})$ to $(1, 20)$ for all datasets.

**Normalization.** In many datasets considered by us, the scale of the time series varies significantly, sometimes by several orders of magnitude. This makes it difficult to train models—particularly those involving neural networks—and the individual time series require normalization before training. Such *per time series* normalization is a standard practice for these datasets and has been employed in several previous works [44, 42, 30]. To this end, we experimented with two normalization methods: standardization and scaling. Given a univariate time series $y_{1:T}$, the normalized version $y_{1:T}^{\text{norm}}$ for each method is computed as

$$\text{Standardization:} \quad y_t^{\text{norm}} = \frac{y_t - \text{mean}(y_{1:T})}{\text{std}(y_{1:T})}, \tag{57}$$

$$\text{Scaling:} \quad y_t^{\text{norm}} = \frac{y_t}{\frac{1}{T} \sum_{i=1}^{T} |y_i|}, \tag{58}$$

where $\text{mean}(\cdot)$ and $\text{std}(\cdot)$ denote the mean and the standard deviation, respectively. The normalization method for each dataset was tuned as a hyperparameter.

**Log-determinant of the Jacobian.** Normalization of data is equivalent to applying a linear transformation to the raw input. More formally, consider a function $f$ that transforms a variable $\mathbf{v} \in \mathbb{R}^D$ to the normalized variable $\mathbf{y} \in \mathbb{R}^D$ via a function $f$, i.e., $f(\mathbf{v}) = \mathbf{y}$. If $p_V(\mathbf{v})$ is the probability density of the random variable $\mathbf{v}$ and $f$ is an invertible and differentiable transformation, then by the change of variables formula the probability density $p_Y(\mathbf{y})$ of the transformed random variable $\mathbf{y}$ is given by

$$p_Y(\mathbf{y}) = p_V(f^{-1}(\mathbf{y})) \left| \det\left( \frac{\partial f^{-1}(\mathbf{y})}{\partial \mathbf{y}} \right) \right|, \tag{59}$$

where $\frac{\partial f^{-1}(\mathbf{y})}{\partial \mathbf{y}} \in \mathbb{R}^{D \times D}$ is the Jacobian of $f^{-1}$. The determinant term accounts for the space distortion of the transformation, i.e., how the volume has changed locally due to the transformation.

Using the change of variables formula, this transformation corresponds naturally to a $\log \det$ term in the log-likelihood which is equal to $J = -\log s$, where $s = \text{std}(y_{1:T})$ in the case of standardization and $s = \sum_{i=1}^{T} |y_i|$ in the case of scaling (the Jacobian in both cases is a diagonal matrix). In the forecasting experiments the $\log \det$ term is included in the objective during training. Note that this

holds only if we treat $s$ as a scaling factor that is independent of the time series $y_{1:T}$; although this is not true, in practice this training heuristic slightly improves the quantitative forecasting performance.

**Number of switches.**   In the case of segmentation, the ground truth number of switches $K$ is known (at least in the context of our experiments). For forecasting, we consider a range of $K \in \{2, \ldots, 5\}$ and we tuned it for each dataset.

**Dimensionality of state $\mathbf{x}_t$.**   We tuned the dimensionality of the state variables $\mathbf{x}_t$ from the set $\{2, 4, 8, 16\}$ for each dataset.

### C.4   Baseline models

In this section, we provide additional details on the baseline forecasting models. The CRPS results for these baseline models have been taken from [30].

**ARSGLS** [30] is an extension of the vanilla SLDS that uses continuous (Gaussian) switch variables. It incorporates conditionally linear state-to-switch recurrence, keeping the conditional tractability of LDS, and augments the emission model by an auxiliary variable that allows for modelling multivariate and non-Gaussian observations with complex non-linear transformations. Inference in ARSGLS is performed via Rao-Blackwellised particle filters wherein the state conditional expectations are computed exactly using the available closed-form expressions whereas the switch and auxiliary variables expectations are approximated via Sequential Monte Carlo using a neural network.

The authors proposed two different instantiations of their model with differences in the underlying base LDSs. In the first instantiation, labelled as RSGLS-ISSM, the authors implemented the LDS as an innovation state space model (ISSM) with a constrained structure that models temporal patterns such as level, trend and seasonality where the transition and emission matrices are pre-defined and not learned. The second instantiation, labelled ARSGLS, uses an unconstrained LDS.

**KVAE** [19] uses a VAE to model auxiliary variables and a "mixture" of LDSs to model the dynamics. KVAE can be interpreted as an SLDS with deterministic switches, where the LDS parameters are predicted as a weighted average of a set of base matrices, with the weights given by an RNN. The RNN in the KVAE depends (autoregressively) on samples of the previous auxiliary variables. Variational inference is performed using a recognition network for the auxiliary variables and Kalman filtering/smoothing for the LDSs' latent variables.

The objective function in KVAE uses samples from the smoothing distribution. However, as noted in [30], the corresponding expectation can be computed in closed-form and the resulting objective function has lower variance. We report the results for both original KVAE (KVAE-MC) and the Rao-Blackwellised variant (KVAE-RB) proposed by Kurle et al. [30].

**DeepState** [42] parametrizes an LDS using an RNN which is conditioned on inputs (controls). The LDS parameters in DeepState have a fixed structure that model time series patterns such as level, trend and seasonality (same as in RSGLS-ISSM). The transition and emission matrices are fixed and the (diagonal) noise covariance matrices are predicted by the RNN directly. Given the LDS parameters from the RNN, DeepState uses the Kalman filter for inference and maximum likelihood for parameter learning.

**DeepAR** [44] is a strong *discriminative* baseline model for probabilistic forecasting that uses an autoregressive RNN conditioned on the history of the time series (lags) and other relevant features. DeepAR autoregressively outputs the parameters of the future distributions and is trained using maximum likelihood estimation.

## D   Computational details

We used a `p3.8xlarge` AWS EC2 instance for running our experiments. This instance comprises 4 Tesla V100 GPUs, 36 CPUs, and 244 GB of memory.