# OpenReview forum: "Deep Explicit Duration Switching Models for Time Series"
_NeurIPS.cc/2021/Conference — NeurIPS 2021 Poster_

### Official Review · Reviewer_Naxb · 2021-07-04

**Rating:** 4
**Confidence:** 3

**Summary:**

The authors proposed a new state-space model which combines the previous two models: recurrent switching LDS and explicit duration together. The authors proposed an inference method to fit the model and applied the proposed method to several real datasets.

**Limitations And Societal Impact:**

See the main review section above for suggestions for improvement. I don't foresee any negative societal impact of their work.

**Main Review:**

Originality: The proposed method combines recurrent state-to-switch with explicit duration. The model is new, but the idea seems straightforward and not revolutionary.

Quality: The paper did extensive experiments on multiple datasets qualitatively and quantitatively. The proposed model has a more complicated structure but doesn't seem to improve previous methods too much. The forecasting results are just comparable with the current methods. The segmentation results will be discussed below.
1. Fig. 3, what are the inputs u for each dataset? Why SNLDS, ED-SDS, RED-SDS got very similar reconstruction, yet different segmentation results?
2. For dancing bee dataset, check paper https://people.eecs.berkeley.edu/~jordan/papers/fox-etal-nips08.pdf. They also fit ARHMM & SLDS with enhanced self-persistence on the dancing bee datasets. The probability distribution of staying at the previous state still follows a geometric distribution. They don't have recurrent state-to-switch or explicit duration in their model, but the results are better than the one shown in the paper.
3. Have you tried a more complicated segmentation task (e.g. number of switching states more than 3, like the one used in https://arxiv.org/pdf/1610.08466.pdf)? Current tasks in the paper are simple and seem not enough to show the superiority of having recurrent state-to-switch and explicit duration. Bouncing ball & 3 mode system are simulated data. Dancing bees data already has good (even better) SOTA results.

Clarity: The paper is clearly written.

Significance: As discussed above, the paper did comparisons with previous methods qualitatively and quantitatively, but doesn't seem to improve current methods too much. The tasks in real data seem not complicated enough. It would be good if the code can be open-source so that others can apply the proposed model to their own datasets.

**Time Spent Reviewing:**

3

---

> ### Author Response · Authors · 2021-08-10
> **Author Response**
>
>
> We thank the reviewer for their comments and suggestions. Please find our responses to specific comments and concerns below:
>
> **Comment**: _“Fig. 3, what are the inputs u for each dataset?”_
> **Response**: We did not use any control inputs for the segmentation experiments (Fig. 3). Control inputs were only available for the forecasting datasets. Please refer to Appendix A.2 for further details.
>
> **Comment**: _“Why SNLDS, ED-SDS, RED-SDS got very similar reconstruction, yet different segmentation results?”_
> **Response**: Achieving good reconstruction results is a common property of models that employ amortized inference. In an amortized inference setup, a neural network maps the input datapoint to the variational distribution instead of using separate variational parameters for each datapoint. This results in an autoencoder-like setup that makes it easy for the model to achieve low reconstruction error without necessarily exhibiting other interpretable properties (such as segmentation and disentanglement).
>
> **Comment**: _Baseline ARHMM & SLDS results in [1] on dancing bees dataset._
> **Response**: Thank you for bringing this paper to our attention. We will cite and discuss it in the revision. Although [1] reports results on the dancing bees dataset, their results are not directly comparable to ours due to differences in the experimental setups. Specifically, we split the long time series into segments of length 120 and used segments from bees 1, 3, 4, 5, 6 for training and segments from bee 2 for testing. This is unlike [1] who did not split the time series. Furthermore, it must be noted that 3 out of 4 methods in [1] (Table 1) namely HDP-VAR(1)-HMM (partially supervised), SLDS DD-MCMC, PS-SLDS DD-MCMC are partially or fully supervised methods, unlike RED-SDS which is completely unsupervised. The only unsupervised method reported in [1], HDP-VAR(1)-HMM, achieves a label accuracy of 44% on bee 2, which although not directly comparable, is significantly lower than our reported value of 73%.
>
> **Comment**: _“...more complicated segmentation task.”_
> **Response**: The primary aim of the segmentation experiments was to demonstrate that the state-of-the-art deep switching model (i.e., SNLDS) is ineffective in modeling time-dependent switching dynamics. Explicit duration modeling improves the capability of the model to capture these time-dependent switching dynamics as shown by the results of ED-SDS and RED-SDS on the 3 mode system and dancing bees datasets.
>
> **Comment**: _“It would be good if the code can be open-source…”_
> **Response**: We have submitted our code as supplementary material and will make it open source after the final notification.
>
> **On quality and significance**
> We reiterate the salient features of this work below:
> - We recognize the inability of state-of-the-art deep switching models to capture time-dependent switching patterns and propose a model with both recurrent state-to-switch connection and explicit durations to handle both state and time-dependent switching. Using carefully designed experiments, we demonstrate scenarios where existing models fail and explicit duration models improve segmentation and interpretability.
> - We propose an efficient inference and learning procedure for the model with succinct gradient computation which results in improved training compared to [2], as also noted by reviewer 2NJA. We will highlight this point further in the revision.
> - In the forecasting experiments, although our model performs the best only on 2/5 datasets, it brings the added benefit of interpretability both in terms of the discrete operating mode and the segment durations. Such insights can be helpful in practical forecasting applications.
>
>
> We thank the reviewer again for their time. We hope that we have satisfactorily addressed the reviewer’s concerns about the significance of this work and the baseline scores on the dancing bees dataset. If so, we hope that the reviewer will consider revising their scores.
>
> [1] Fox, Emily, et al. "Nonparametric Bayesian learning of switching linear dynamical systems." Advances in Neural Information Processing Systems 21 (2008): 457-464.
> [2] Dong, Zhe, et al. "Collapsed amortized variational inference for switching nonlinear dynamical systems." International Conference on Machine Learning. PMLR, 2020.

---

### Official Review · Reviewer_2NJA · 2021-07-14

**Rating:** 7
**Confidence:** 4

**Summary:**

In this paper, the authors introduce RED-SDS an extension to switching dynamical systems (SDS) that combines state-dependent and duration-dependent transitions.

**Limitations And Societal Impact:**

The authors adequately addressed the limitations and potential negative societal impact of their work.

**Main Review:**

# Strengths
I think the combination of state-dependent and duration-dependent transitions is very cool, as it leads to a much more flexible model. For learning RED-SDS, the authors build upon previous work on variational inference for SDS [1]. Crucially, the authors demonstrate the gradient for$ \underset{q(x_{1:T} \vert y_{1:T})}{\mathbb{E}}  [\log p(y_{1:T}, x_{1:T})]$ can be computed more succinctly as opposed to [1]; moreover, this also leads to better training, removing the need of the regularizer that was used in [1]. The experiments section was also very well done and very thorough.

# Weaknesses
I noticed that details are missing for the experiments section. In neither the main text nor the appendix, are there details regarding the architectures being used i.e. number of hidden units and activation functions. Moreover, I don't see any details regarding the annealing schedule for the tempered softmax function.

# Questions/Comments
1) In line 556, shouldn't it be $ P \in \mathbb{R}^{K \times (d_{max} - d_{min})}$, not $ P \in \mathbb{R}^{K \times d_{max}}$?
2) In line 602, since the latent dimension is 2, I don't think it qualifies as a univariate time series.
3) In lines 619-629, the dancing bees dataset was described where it was stated that the data is comprised of 6 bees. Were the methods only trained on data from one bee or where each bee considered iid?
4) In line 671, it was stated the dimensionality of the state variables was set to 4 for the bouncing ball and 3 mode datasets. I find this weird (and slightly concerning), as 1) the true latent dimension is known and 2) the data generating process is a subset of RED-SDS, thus setting the latent dimension to be the true dimension should suffice.
5) In lines 769-773, the authors noted that they took the scaling of the data into effect by using the change of variables formula. This is interesting because the standard is just to use scale the data and treat it as if it wasn't scaled. I'm curious as to why the authors did this (Note, that the change of variables formula is only for one-to-one function; the sample average and standard deviation are not one-to-one).
6) In line 776, the authors stated they considered a range of $K \in [2, 5]$. I think it would be more appropriate to write it as $K \in$ {$2, 3, 4, 5$}, as $K$ can only be discrete.

# Conclusion
I think this was a good paper overall! Only minor modifications are needed.

# References
[1] Collapsed amortized variational inference for switching nonlinear dynamical systems, https://arxiv.org/abs/1910.09588

**Time Spent Reviewing:**

10 hours

---

> ### Author Response · Authors · 2021-08-10
> **Author Response**
>
> We thank the reviewer for their positive comments and the thoroughness of their review. We especially appreciate the comments on the succinct gradient computation of $\mathbb{E}_{q(\mathbf{x}\_{1:T}|\mathbf{y}\_{1:T})}[\log p(\mathbf{x}\_{1:T}, \mathbf{y}\_{1:T})]$ which results in a straightforward implementation (using only the last filtered distribution) and improved training. Please see our response to specific comments and questions below:
>
> **Comment**: _Missing architecture and annealing details._
> **Response**: We will add these details to the appendix. Specific experiment details can also be found in the accompanying code. We will also open source our implementation with experiment configs after the final notification.
>
> **Comment**: _Should the duration matrix be $\mathbf{P} \in \mathbb{R}^{K \times (d\_{\mathrm{max}} - d\_{\mathrm{min}})}$?_
> **Response**: Thank you for spotting this error. Indeed it should be $\mathbf{P} \in \mathbb{R}^{K \times (d_{\mathrm{max}} - d_{\mathrm{min}})}$. We will fix this in the revision.
>
> **Comment**: _“In line 602, since the latent dimension is 2, I don't think it qualifies as a univariate time series.”_
> **Response**: Although the latent dimension is 2, the observations are 1-dimensional, hence the term univariate. We will clarify this term in the text to help other readers with a similar concern.
>
> **Comment**: _“...Were the methods only trained on data from one bee or where each bee considered iid?”_
> **Response**: As mentioned in lines 626-629, we split the long time-series into chunks of 120 timesteps. We used the resulting chunks from bees 1, 3, 4, 5, 6 for training, treating them as iid datapoints.
>
> **Comment**: _Why was the dimensionality of state variables set to 4 for the bouncing ball and 3 mode datasets?_
> **Response**: In this work, we focused on interpretable discrete switch dimensions; therefore, we set the number of switches $K$ equal to the known ground truth value for both datasets. For the dimensionality of state $\mathbf{x}_t$, we set it to 4 for learning and inference flexibility. Note that this is because we perform amortized inference for the state variable $\mathbf{x}_t$ via a recognition network and not exact inference. Eventually, if the dimensionality proves to be more than required for modeling the data, the neural network ignores the extra dimensions or learns redundant information in them. This does not hamper learning or inference in any way. Moreover, the true dimensionality of the state $\mathbf{x}_t$ is rarely known in practice. We will add these clarifications to the appendix.
>
> **Comment**: _On scaling and the change of variables formula._
> **Response**: The individual time series in the datasets we have considered vary significantly in scale. Therefore, we perform normalization/standardization _per time series_ rather than across the dataset. Note that such per time series normalization is a standard for these datasets and has been employed in several previous works (e.g., [1, 2, 3]).
>
> Regarding the change of variables, once the scale value has been precomputed for each time series in the dataset, the actual scaling operation only amounts to dividing the observation at each time step by a scalar value. This results in a bijective map where the change of variables formula is applicable. The log-det term takes into account the relative scale of the different time series and their relative contribution to the conditional likelihood term in the loss. This log-det term is also used in the [GluonTS](https://github.com/awslabs/gluon-ts) time series library for models such as DeepAR [1]. In our experience, the addition of the log-det term only slightly improves the quantitative forecasting performance and is not essential for effective training of our proposed model. Given the question of the reviewer, we will add these details to the appendix.
>
> **Comment**: _Notation for the set of K values._
> **Response**: We will fix this typographical error.
>
> We thank the reviewer again for their positive comments and hope that we have satisfactorily answered their questions.
>
> [1] Salinas, David, et al. "DeepAR: Probabilistic forecasting with autoregressive recurrent networks." International Journal of Forecasting 36.3 (2020): 1181-1191.
> [2] Kurle, Richard, et al. "Deep Rao-Blackwellised particle filters for time series forecasting." Advances in Neural Information Processing Systems 33 (2020).
> [3] Rangapuram, Syama Sundar, et al. "Deep state space models for time series forecasting." Advances in neural information processing systems 31 (2018): 7785-7794.

---

> > ### Comment · Reviewer_2NJA · 2021-08-30
> > **Response to the authors**
> >
> > Thanks so much for responding to my comments. Most of my concerns have been addressed. I just have a few lingering questions:
> >
> > 1) I'm confused about the response to my comment about dimensionality for the bouncing balls and 3 mode systems. Specifically, the authors stated that they set the number of discrete states $K$ to be the true value, why not do the same for the latent dimension. As REDS is a generalization of the data generating process for bouncing balls and 3 mode systems, it should be able to perform learning and inference using the true latent dimension, no?
> >
> > 2) I want to clarify my point concerning the scaling of the data. I agree for a fixed and constant location and scaling term, the change of variables formula can be straightforwardly applied *but* in this setting, the location and scale are a function of data hence the change of variables formula isn't appropriate in this setting.

---

> > > ### Author Response · Authors · 2021-08-31
> > > **Author response to follow-ups**
> > >
> > > Thank you for your response. Please find our responses to your comments below:
> > >
> > > **Comment**: _On the dimensionality of the state_ $\mathbf{x}_t$.
> > > **Response**: We agree that the model should be able to perform well with the true state dimension; however, we did not perform experiments with these settings as the interpretability of state dimension was not the focus of our experiment. Our primary focus for the segmentation experiment was the interpretability of the switch variable; we set the value of $K$ equal to the ground truth and chose a reasonable value for the state dimensionality, depending on the complexity of the dataset and to facilitate flexibility in amortized inference (note that amortized inference was only performed for the state and not the switch variables). We thank the reviewer for the comment and we have kicked off additional experiments with the true state dimension. We will include the results in the revision and add preliminary results here if they finish before the discussion period ends.
> > >
> > > **Comment**: _On the change of variables formula._
> > > **Response**: The scaling terms for the time series are precomputed and are available as scalars per time series. Note that these terms are not computed via a differentiable computation graph, which would be incorrect as the reviewer has rightly pointed out. We would like to reiterate that the log-det term is only a (non-essential) training heuristic that slightly improves the forecasting performance when the scales of the time series in the data vary significantly.
> > >
> > > Thank you again for your insightful comments and suggestions.

---

> > > > ### Comment · Reviewer_2NJA · 2021-08-31
> > > > **response**
> > > >
> > > > Thanks for the clarifcations!

---

> > > ### Author Response · Authors · 2021-09-01
> > > **Preliminary results**
> > >
> > > Preliminary results on the bouncing ball dataset with `x_dim=2` for RED-SDS are tabulated below along with the results reported in the paper (with `x_dim=4`).
> > >
> > > ```
> > > +----------+-------------------------+-------------------------+
> > > |    --    | Bouncing Ball (x_dim=4) | Bouncing Ball (x_dim=2) |
> > > +----------+-------------------------+-------------------------+
> > > | Accuracy | 0.97 (0.00)             |    0.97 (0.00)          |
> > > | NMI      | 0.81 (0.00)             |    0.80 (0.01)          |
> > > | ARI      | 0.88 (0.00)             |    0.88 (0.00)          |
> > > +----------+-------------------------+-------------------------+
> > > ```
> > >
> > >
> > > In this experiment, we kept all the tuned hyperparameters from the original experiment fixed and only changed `x_dim` to 2. Notably, the model with `x_dim=2` is able to retain the same performance as that reported in the paper. These results support our hypothesis that the model should be able to perform well when given the true state dimensionality. In practice, `x_dim` should be greater than or equal to the ground truth for the model to perform well.
> > >
> > > We are conducting similar experiments for the 3 mode system dataset and will report the results for the different model variants (SNLDS, ED-SDS, and RED-SDS) on both the datasets in the revision.

---

> > > > ### Comment · Reviewer_2NJA · 2021-09-01
> > > > **response**
> > > >
> > > > This is great to see! Thank you!

---

### Official Review · Reviewer_ByG2 · 2021-07-16

**Rating:** 5
**Confidence:** 3

**Summary:**

The authors propose Recurrent Explicit Duration Switching Dynamical System model, that is used to detect state and time switching dynamics. The method falls to the category of generalized switching state space model (Ghahramani & Hinton 2000). In particular, they use recurrent state-to-state switching with explicit duration models and provide approximate inference algorithm that is using evidence lower bound. Authors verify they model on 3 segmentation task from time series with ground truth and forecasting task on 5 datasets.

**Ethical Concerns:**

no ethical concerns.

**Ethics Review Area:**

["I don’t know"]

**Limitations And Societal Impact:**

yes.

**Main Review:**

Authors make a nice study, however I am not sure about the following:
1. Related work on jump models
The authors are missing big part of related work related to Jump models. Jump models have been intensively used in financial machine learning and they also generalize hidden markov models by having abrubt and presistent change of state dynamics.
See Bemporad et al. (2018), that has proved that the HMM is a special case of a jump model.

Bemporad, A., V. Breschi, D. Piga, and S. Boyd. “Fitting jump models.” Automatica,
vol. 96 (2018), pp. 11–21.

2. How does their model tolerate more noisy datasets
Authors should try to see how they model behaves on very noisy datasets e.g. financial.

3. Based on the results in table 2 for the forecasting, the method is performing the best only on 2 datasets. Do authors have intuition why?

**Time Spent Reviewing:**

5

---

> ### Author Response · Authors · 2021-08-10
> **Author Response**
>
> Thank you to the reviewer for their time and comments. Our responses to specific comments and questions are as follows:
>
> **Comment**: _On jump models and financial datasets._
> **Response**: We thank the reviewer for their pointers to jump models, which we will cite accordingly in the revised version of the paper. However, we would like to point out that our work differs from jump models in several respects: our model is built on a probabilistic framework and is mainly concerned with forecast distributions. Moreover, it also focuses mainly on employing neural networks to approximate the data generating process.
>
> We would also like to point out that the first data set (“exchange”) the model is tested on is a daily exchange rate data set over 8 currency pairs, and our model is shown to provide the most favorable results. We concur with the reviewer however that financial econometrics are an interesting domain and that more extensive experiments in a follow up paper should focus on this. Since we had a general forecasting problem set-up in mind for the present paper, we did not straight-away consider these beyond the exchange data set.
>
> **Comment**: _Why does the model perform best on only two forecasting datasets?_
> **Response**: The datasets considered for the forecasting experiment are diverse and exhibit different domains, frequencies, trend and seasonality patterns. Thus, it is difficult to hypothesize why the model performs the way it does on the individual datasets. It is also noteworthy that no baseline method consistently performs best on all the datasets.

---

### Author Response · Authors · 2021-08-26
**Final Author Remarks**

We would like to thank the reviewers again for their time and comments. As the discussion period is almost over, we hope that we have adequately answered all the reviewers' comments. If further clarifications are needed, please feel free to let us know.

---

### Decision · Program_Chairs · 2021-09-27

**Decision:**

Accept (Poster)

**Comment:**

This paper takes a nice idea (combining recurrent state-to-switch dependencies with semi-Markov duration distributions), pairs it with an amortized inference algorithm, describes the approach very well, and does a thorough set of experiments with reasonable baseline comparisons to recent methods. While it is a relatively simple idea, it is well executed and I think it will be a very nice contribution to the field.

In private discussion, Reviewers Naxb and GyB2 agreed that they would increase their scores to a 6, which would yield two 6s and a 7, for an average of 6.33.  I strongly encourage the authors to make the promised changes in the final publication.